# BRD4-mediated repression of p53 is a target for combination therapy in AML

Anne-Louise Latif[1,10], Ashley Newcombe[1,10], Sha Li [2], Kathryn Gilroy[1], Neil A. Robertson [1], Xue Lei[2], Helen J. S. Stewart [3], John Cole[1], Maria Terradas Terradas [1], Loveena Rishi[1], Lynn McGarry [4], Claire McKeeve[5], Claire Reid[1], William Clark[4], Joana Campos[6], Kristina Kirschner [1], Andrew Davis[2], Jonathan Lopez[1], Jun-ichi Sakamaki[4], Jennifer P. Morton [1,4], Kevin M. Ryan[4], Stephen W. G. Tait [1], Sheela A. Abraham [6,8], Tessa Holyoake[6], Brian Higgins[7], Xu Huang [6], Karen Blyth [1,4], Mhairi Copland [6], Timothy J. T. Chevassut [3], Karen Keeshan [6] & Peter D. Adams [1,9✉]

Acute myeloid leukemia (AML) is a typically lethal molecularly heterogeneous disease, with few broad-spectrum therapeutic targets. Unusually, most AML retain wild-type *TP53*, encoding the pro-apoptotic tumor suppressor p53. MDM2 inhibitors (MDM2i), which activate wild-type p53, and BET inhibitors (BETi), targeting the BET-family co-activator BRD4, both show encouraging pre-clinical activity, but limited clinical activity as single agents. Here, we report enhanced toxicity of combined MDM2i and BETi towards AML cell lines, primary human blasts and mouse models, resulting from BETi's ability to evict an unexpected repressive form of BRD4 from p53 target genes, and hence potentiate MDM2i-induced p53 activation. These results indicate that wild-type *TP53* and a transcriptional repressor function of BRD4 together represent a potential broad-spectrum synthetic therapeutic vulnerability for AML.

[1] Institute of Cancer Sciences, University of Glasgow, Glasgow, UK. [2] Sanford Burnham Prebys Medical Discovery Institute, San Diego, CA, USA. [3] Brighton and Sussex Medical School, University of Sussex, Brighton, UK. [4] Cancer Research UK Beatson Institute, Glasgow, UK. [5] West of Scotland Genomics Services (Laboratories), Queen Elizabeth University Hospital, Glasgow, UK. [6] Paul O'Gorman Leukemia Research Centre, Institute of Cancer Sciences, College of Medical Veterinary and Life Sciences, University of Glasgow, Glasgow, UK. [7] Pharma Research and Early Development, Roche Innovation Center-New York, New York, USA. [8] Present address: Department Of Biomedical And Molecular Sciences, Queen's University, Kingston, Ontario, Canada. [9] Present address: Sanford Burnham Prebys Medical Discovery Institute, San Diego, CA, USA. [10] These authors contributed equally: Anne-Louise Latif, Ashley Newcombe. ✉email: padams@sbpdiscovery.org

D espite numerous advances in the knowledge of the molecular landscape of Acute Myeloid Leukemia (AML), there remains an unmet need to improve clinical outcomes; 5-year survival rates in 2019 for adults diagnosed with AML remain below 30%[1]. AML is a genetically and epigenetically heterogeneous disease, characterized by recurrent but diverse chromosomal structural changes and genetic mutations associated with functionally distinct sub-groups[2,3]. In addition to disease heterogeneity between patients, marked sub-clonal heterogeneity has also been observed within individual AML patients[4]. To date these disease features have posed a significant obstacle to finding novel targeted agents with a broad therapeutic reach.

One key unifying feature of AML, that could potentially be exploited to benefit many patients, is that the majority of cases exhibit wild-type *TP53*[2,3]. In AML with wild-type *TP53*, the p53 tumour suppressor protein is commonly held functionally inert through dysregulation of the ARF-MDM2/4 axis, culminating in inactivation of p53 by its negative regulators, MDM2 and MDM4[5]. Drugs and small molecules have been developed that can activate p53 in cells expressing the wild-type gene[6], with the goal of unleashing p53's potent tumor suppressive functions. Clinical grade MDM2 inhibitors (MDM2i) have been tested in the clinic, for example RG7112 in hematological and solid tumors, with some encouraging, but limited, responses[6,7]. Nevertheless, in pre-clinical studies, MDM2i cooperated with "standard-of-care" therapies, daunorubicin and cytarabine, to eradicate AML[8]; and the BCL2 inhibitor (Venetoclax) and MDM2i (Idasanutlin/RG7388 [a clinical grade RG7112 derivative with improved potency, selectivity and bioavailability][9]) are also synthetic lethal in AML[10]. Other combination strategies have also demonstrated the utility of targeting wild-type p53 in AML[11]. Consistent with these studies, Andreeff *et al* have proposed that use of MDM2i to activate p53 will likely realize more benefit in combination therapies[7].

Bromodomain-containing protein 4 (BRD4) is a member of the bromodomain and extraterminal (BET) family proteins, characterized by two N-terminal bromodomains and an extraterminal domain[12]. BRD4 has been shown to play a role in the activation of genes involved in cell growth - most notably *c-MYC* - through binding to acetylated histones and transcription factors, to which BRD4 then recruits transcriptional regulators, such as positive transcription elongation factor b (P-TEFb) and Mediator complex[12]. Although *c-MYC* translocations or mutations are not common in AML, the activation of *c-MYC* by multiple up-stream leukemic genetic aberrations has been recognized as a key hub in driving leukemogenesis[13]. Pre-clinical data has demonstrated that inhibition of BRD4 has efficacy across a range of AML subtypes[14–16]. Indeed, BET inhibitors (BETi) have entered early phase clinical trials for AML. However, despite promising pre-clinical activity, their efficacy in treating AML as single agents has been modest[17–20], and as such it is likely that, like MDM2i, their strength lies in rational combination therapies.

In sum, both MDM2i and BETi have been considered as therapies for AML, but on their own have shown limited clinical activity[6,7,17–20]. Given that both drugs can, in principle, target a broad spectrum of AML molecular subtypes and the two drugs have distinct modes of action, we set out to test the hypothesis that the concomitant reactivation of p53 and inhibition of BET family proteins, using MDM2i and BETi, could synergise to kill AML cells. Here we present data showing superior efficacy of the drug combination over the single agents in genetically heterogenous AML cell lines, primary AML samples, and two relevant mouse models. We present mechanistic data demonstrating how this efficacious drug combination co-operates to induce pro-apoptotic p53 target genes.

## Results

**BETi enhance the killing of human AML cells by MDM2i in a p53-dependent manner.** Initial experiments to assess potential synergy of the MDM2i and BETi combination were performed in a panel of primary AML cells from 15 heterogeneous AML patients. These patients had a median age of diagnosis of 60 years (range 31 to 78 years). Based on their non-complex karyotype we expect the majority to retain wild-type *TP53*[2], and this was confirmed for the 4 samples for which DNA sequencing data was available (Supplementary Fig. 1A). In these initial in vitro studies, for BETi we used CPI203 (Constellation Pharmaceuticals), and for MDM2i we used nutlin-3 (Sigma). CPI203 is a potent BETi with an IC50 of 37 nM for inhibition of BRD4[21]. Like other BETi, CPI203 represses expression of oncoproteins, such as c-MYC[21]. CPI203 is a pre-clinical tool compound version of CPI0610, another Constellation BETi already being tested in human AML[22]. Nutlin-3 is a potent inhibitor of the interaction between p53 and its negative regulator MDM2[23], thereby activating p53 to inhibit cell proliferation and tumorigenesis in models harboring wild-type *TP53*. As an assessment of potential drug synergy, we measured cell viability reflected in ATP levels (CellTitre-Glo, Promega). Across all 15 patient samples, the combination of CPI203 and nutlin-3 was significantly more efficacious than either drug alone (Fig. 1a). To more quantitatively assess drug interactions, we calculated the coefficient of drug interaction (CDI) for each sample; CDI < 1, =1 or >1 indicates that the drugs are synergistic, additive or antagonistic, respectively. The majority of the samples showed a trend towards greater than additive killing by the combination (x axis < 1), compared to the single drugs (Fig. 1b). For 4 of the samples, this synergistic interaction was determined to be significant (p < 0.05). Due to limited sample availability, *TP53* mutation status could only be determined by DNA sequencing for 4 of the human patient samples. Of these, all retained wild-type *TP53* and 3 of them showed at least a trend towards synergy. These results in primary patient samples point to enhanced toxicity of the BETi and MDM2i combination, compared to single agents, against a substantial proportion of primary human AML.

To extend these results obtained in primary AML blasts and to be able to perform more rigorous mechanistic analyses, we tested synergy between nutlin-3 and CPI203 in a panel of 3 human AML cell lines expressing wild-type p53 (OCI-AML3, MOLM13, MV411). To begin to assess the p53 dependence of any effect, we also tested 3 cell lines with mutant *TP53* (THP1, KG1a, Kasumi1). Irrespective of *TP53* status, these AML cell lines reflect diverse AML subtypes. For example, OCI-AML3 harbor *DNMT3A* and *NPM1* mutations, MOLM13 and MV411 both contain mutant *FLT3-ITD*, MOLM13 expresses an MLL-AF9 fusion oncoprotein, while MV411 expresses the MLL-AF4 fusion oncoprotein (ref. [24] and ATCC). We confirmed initially that the drugs engaged their molecular targets, reflected in upregulation of p53 and CDKN1A (p21) by nutlin-3, and down regulation of c-MYC by CPI203 (irrespective of *TP53* status) (Fig. 1c, d). To compare the effects of the single drugs and combination on viability of these 6 cell lines, we used resazurin (a metabolic viability sensor) across a range of fixed drug dose ratios, enabling drug interaction to be quantitatively assessed by the combination index (CI) method, where CI < 1 is generally considered as synergy[25]. Consistent with previous studies in OCI-AML3 cells[26], the drug combination demonstrated apparent synergy in wild-type *TP53* cell lines (Fig. 1e–g and Supplementary Fig. 1B–D). In the *TP53* wild-type OCI-AML3 cell line, CPI203 and nutlin-3 were designated synergistic from CI 0.34–0.52 at different drug ratios (Fig. 1e and Supplementary Fig. 1B–D). Similarly, MV411 and MOLM13-showed CI 0.21 and 0.29, respectively (Fig. 1f, g). There was no benefit in using the combination treatment on the *TP53* mutated

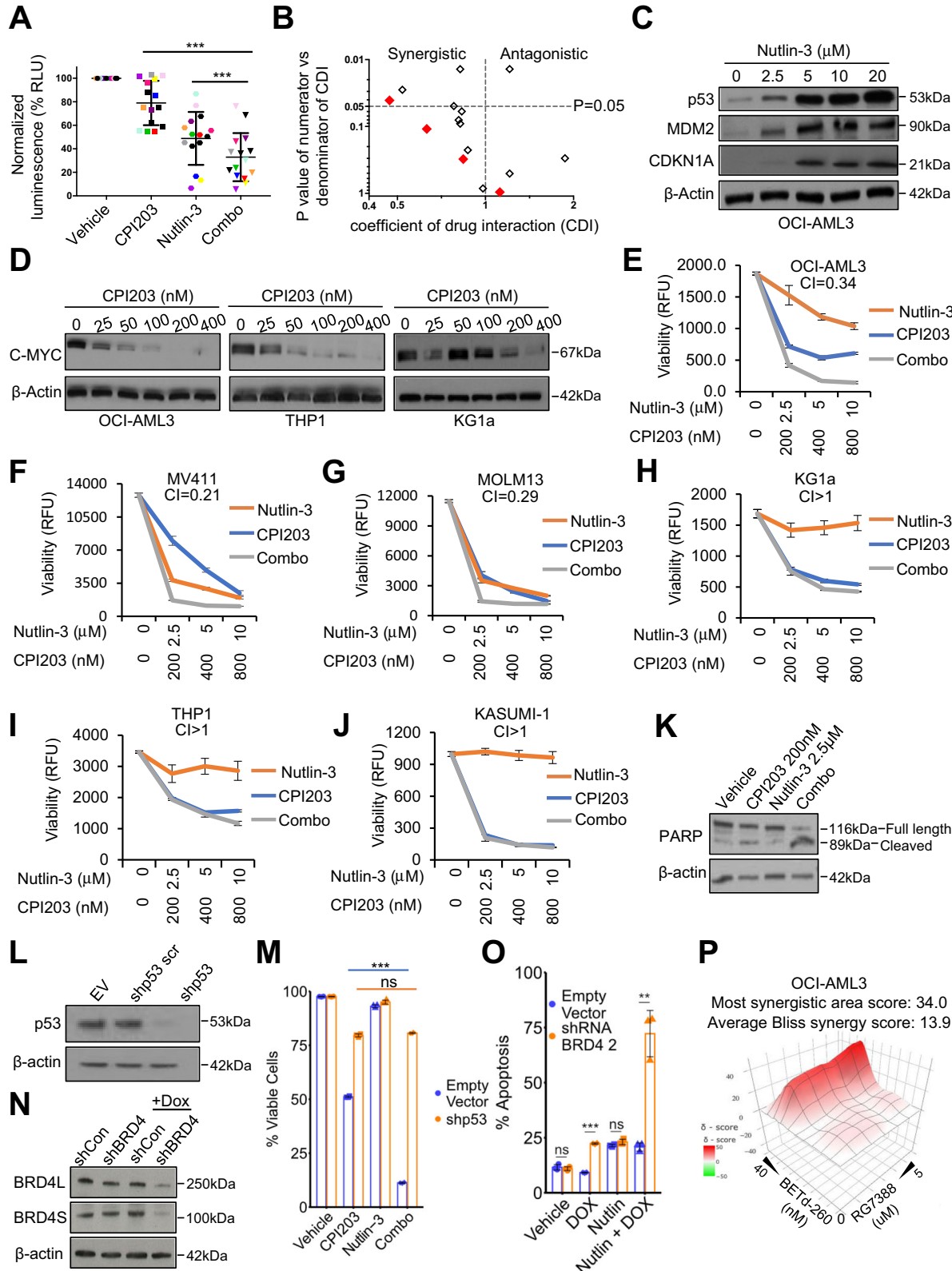

cell lines (Fig. 1h–j) (CI for all mutant *TP53* lines > 1). To confirm these results we also calculated drug synergy by the Bliss method[27]. This also showed substantially greater and more consistent synergy (Bliss score >0) of the combination in *TP53* wild-type cells, across a range of drug doses (Supplementary Fig. 1E–J). The results of the cell viability studies in the *TP53* wild-type OCI-AML3 cell line were corroborated by annexin

V/propidium iodide flow cytometry assay. We observed marked enhancement of apoptosis specifically (annexin V positive) and/ or cell death by apoptosis or other mechanisms (annexin V/PI positive), compared to either single drug alone (Supplementary Fig. 1K–M). Enhanced killing of the combination-treated cells was accompanied by a clear increase in cleaved PARP (Fig. 1k), a marker of pro-apoptotic caspase activation, showing that the drug

**Fig. 1 MDM2 and BET inhibitors combine to enhance killing of primary human AML blasts and AML cell lines with wild-type *TP53*. a** Primary human patient AML blasts (n = 15) were treated with indicated drugs (2.5 μM nutlin-3 alone, 200 nM CPI203 alone, and the two in combination) and cell viability was assessed by CellTiter-Glo assay after 48 h (***=$p \leq 0.001$, two-tailed unpaired t-test). **b** Scatter plot of coefficient of drug interaction (CDI) and synergy significance in primary human patient blasts from A. CDI was calculated as follows: CDI = AB/(AxB). AB is the fraction of cells surviving (0–1.0) in the combination of drugs (nutlin-3 + CPI203); A (nutlin-3) and B (CPI203) are the fraction of cells surviving in each of the single drugs. CDI value (x-axis) <1 (left), =1, or >1 (right) indicates that the drugs are synergistic, additive or antagonistic, respectively. p value (y-axis) ≤0.05 indicates significance (top). The *p* value was generated comparing AB and AxB (***=$p \leq 0.001$, two-tailed unpaired t-test). Red diamonds, *TP53* wild-type. Open diamonds, *TP53* status unknown. **c** Western blots performed on the OCI-AML3 cell lines assessing expression of p53, MDM2, and CDKN1A after 24 h of drug treatment with increasing doses of nutlin-3. **d** Western blots performed on the OCI-AML3, THP1, and KG1a cell lines, assessing the expression of C-MYC after 24 h in indicated doses of CPI203. **e** OCI-AML3 cell viability (each treatment in triplicate) was assessed by resazurin assay after 72 h, using a treatment ratio of CPI203:nutlin-3 of 1:12.5. Mean ± Standard deviation of 3 independent replicates is shown. **f** MV411 cell viability (each treatment in triplicate) was assessed by resazurin assay after 72 h, using a treatment ratio of CPI203:nutlin-3 of 1:12.5. Mean ± Standard deviation of 3 independent replicates is shown. **g** MOLM13 cell viability (each treatment in triplicate) was assessed by resazurin assay after 72 h, using a treatment ratio of CPI203:nutlin-3 of 1:12.5. Mean ± Standard deviation of 3 independent replicates is shown. **h** KG1a cell viability (each treatment in triplicate) was assessed by resazurin assay after 72 h, using a treatment ratio of CPI203:nutlin-3 of 1:12.5. Mean ± Standard deviation of 3 independent replicates is shown. **i** THP1 cell viability (each treatment in triplicate) was assessed by resazurin assay after 72 h, using a treatment ratio of CPI203:nutlin-3 of 1:12.5. Mean ± Standard deviation of 3 independent replicates is shown. **j** KASUMI-1 cell viability (each treatment in triplicate) was assessed by resazurin assay after 72 h, using a treatment ratio of CPI203: nutlin-3 of 1:12.5. Mean + /- Standard deviation of 3 independent replicates is shown. **k** Cleavage of PARP as assessed by Western blot analysis according to treatment condition after 24 h treatment of the OCI-AML3 cell line. **l** Western blot assessing expression of p53 in the OCI-AML3 cell line, harboring empty vector, scrambled shRNA p53 or shRNA p53. **m** Cell viability as assessed by trypan blue in OCI-AML3 cells harboring empty vector or shRNA p53, treated for 72 h with vehicle, 200 nM CPI203, 2.5 μM nutlin-3 or the drug combination (***=$p \leq 0.001$, two-tailed unpaired t-test, n = 3, Means ± SD are shown). **n** Western blot analysis of shRNA-mediated knockdown of BRD4 (shRNA induced by 0.5ug/ml doxycycline for 72 h). **o** FACS analysis of apoptosis (annexin V and PI) in OCI-AML3 cells expressing empty-vector or shRNA BRD4 (doxycycline-inducible), treated with vehicle, doxycycline, nutlin-3, and the combination of nutlin-3 and doxycycline for 72 h (***=$p \leq 0.001$, **=$p \leq 0.01$, two-tailed unpaired t-test, n = 3, Means ± SD are shown). **p** Excess Over Bliss plot showing synergistic effects between RG7388 and BETd-260 in OCI-AML3 cells. Cell viability (each treatment in quadruplicate) was assessed by CellTiter-Glo after 24 h. Bliss synergy scores were indicated.

combination induces apoptosis in these *TP53* wild-type AML cells.

Confirming a requirement for wild-type *TP53* for the drug combination to induce super-additive cell death, we knocked down p53 using shRNA in the *TP53* wild-type OCI-AML3 cell line which abrogated the benefit of the drug combination (Fig. 1l, m). To similarly confirm that the effects of CPI203 were mediated via inhibition of a BET family member, we asked whether the enhanced killing by the MDM2i/BETi combination was recapitulated by MDM2i together with BRD4 knockdown. Indeed, inducible shRNA-mediated knockdown of BRD4 markedly potentiated killing by MDM2i (Fig. 1n, o). In addition, we used BET PROTACs to selectively and completely induce degradation of BRD4 in cells (Supplementary Fig. 1N)[28,29]. Consistent with BETi, BET degraders (BETd-260 and ARV-825) synergize with MDM2i to suppress cell viability in 3 wild-type *TP53* cell lines (OCI-AML3, MOLM13, MV411) (Fig. 1p and Supplementary Fig. 1O–S). Taken together, in primary human AML blasts and human AML cell lines, we conclude that, in AML expressing wild-type p53, BETi inhibit BRD4 to enhance cell killing by MDM2i.

**BETi and MDM2i cooperate to eradicate AML in mouse models.** Next, we asked whether the drug combination shows superior anti-leukemic activity to the single agents, in two murine models of AML. *Trib2* is an oncogene capable of causing AML in mice through down-regulation of the transcription factor C/EBPα, a gene that is mutated in 15% of cases of AML in humans[30]. Following confirmation by PCR and Sanger sequencing that blasts from mice with AML driven by Trib2 expressed wild-type p53 (data not shown), we first confirmed superior eradication of the murine leukemic cells by the drug combination compared to the two single drugs in vitro (Fig. 2a–c), recapitulating the effect in human cells. For the purposes of subsequent in vivo work we wanted to use clinical-grade drugs. We used the clinical-grade BETi CPI0610 (Constellation Pharmaceuticals) and MDM2i RG7112 (Roche Pharmaceuticals). RG7112 has been

tested as a single agent in relapsed refractory AML[7], showing modest activity, and is better suited to studies with murine AML than human-optimized RG7388[9]. CPI0610 is currently being tested in early-stage human clinical trials and has been reported to be well-tolerated[31]. For our experiments, we also identified well-tolerated drug doses in mice, based on a 21-day drug pilot experiment, assessing weight loss and myeloid cell counts (in bone marrow), B cells (in spleen), and T cells (in thymus) in non-leukemic normal healthy mice (Supplementary Fig. 2A, B). A total of 40 C57BL/6 mice (10 mice each for vehicle, both single drugs and the combination) were sub-lethally irradiated and $0.85 \times 10^6$ Trib2 AML blasts injected via their tail veins. Myeloblasts expressed GFP from the same retroviral construct as Trib2 for disease tracking. 21 days of drug treatment was commenced in all leukemic mice, post confirmation of comparable disease engraftment between groups (data not shown). Three mice from each treatment group were sacrificed after the first 48 h of drug treatment, GFP+ blasts recovered from bone marrow, RNA extracted and qPCR performed to demonstrate that RG7112 increased expression of *CDKN1A* and CPI0610 reduced levels of *c-MYC* (Fig. 2d, e), as expected if the drugs engage their respective targets. After 21 days of treatment, all remaining mice were sacrificed (n = 7 for each group, except n = 6 for vehicle because 1 succumbed to disease 15 days post engraftment) and abundance of GFP+ AML blasts in bone marrow determined by FACS. The drug combination demonstrated significantly enhanced anti-leukemic activity compared to either single drug alone, measured in bone marrow (Fig. 2f, g), spleen, and thymus (Supplementary Fig. 2C). In bone marrow and spleen the CDI was <1 (0.017 and 0.38 respectively), indicating synergy in vivo.

Following demonstration that the drug combination was tolerable and efficacious at the end of the treatment period, we assessed if the drug combination could confer a survival advantage over the single agents in mice with Trib2-driven AML. Sub-lethally irradiated cohorts of mice were again injected with GFP+ Trib2 AML blasts and, following comparable disease engraftment between groups (Supplementary Fig. 2D), were

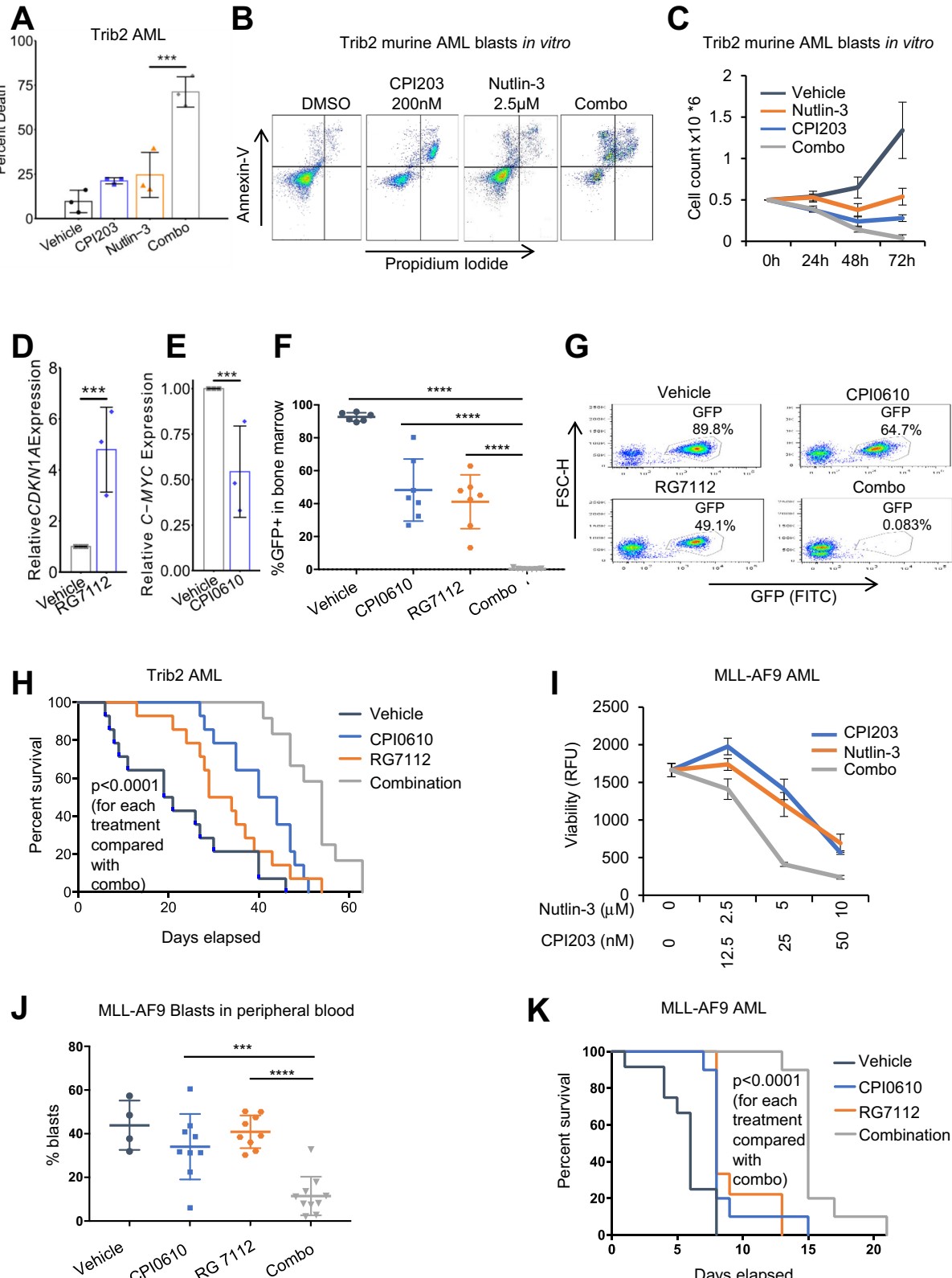

treated with vehicle, single drugs, or combination for 21 days. FACS analysis of peripheral blood at the end of drug treatment confirmed the synergy of the drug combination in eradication of GFP+ blasts (CDI = 0.031, Supplementary Fig. 2E). After completion of the 21-day drug cycle, mice were closely monitored for signs of leukemia and culled at an ethical endpoint. Although

the combination-treated mice did ultimately succumb to disease after cessation of treatment, indicative of low-level residual disease, the combination-treated mice did have a statistically significant survival advantage compared with the single agent-treated mice (Fig. 2h). This underscores the anti-leukemic activity of the combination in this model.

**Fig. 2 MDM2 and BET inhibitors cooperate to eradicate AML in in vivo mouse models. a** Primary murine Trib2 AML blasts, treated in vitro for 72 h with vehicle, 200 nM CPI203, 2.5 μM nutlin-3, or the drug combination. Cell death assessed by flow cytometry using Annexin-V and PI staining (***=$p \leq 0.001$, two-tailed unpaired t-test, $n = 3$, Means ± SD are shown). **b** A representative example of the data summarized in Fig. 2a (Gating strategy for sorting GFP positive cells was live cells>cell profile>single cells>GFP positive cells). **c** A time course of viable cell counts as assessed by trypan blue staining, of primary murine Trib2 AML blasts, according to treatment condition (Error bars are SEM, $n = 3$, Means ± SD are shown). **d** qPCR assessment of CDKN1A expression in GFP+ blasts recovered from bone marrow from three mice sacrificed after 48 h treatment with RG7112 relative to vehicle (**=$p \leq 0.01$, two-tailed unpaired t-test, $n = 3$, Means ± SD are shown). **e** qPCR assessment of C-MYC expression in GFP+ blasts recovered from bone marrow from three mice sacrificed after 48 h treatment with CPI0610 relative to vehicle (***=$p \leq 0.001$, two-tailed unpaired t-test, $n = 3$, Means ± SD are shown). **f** Assessment of the disease burden in the bone marrow according to treatment condition, at the end of 21 days of treatment in vivo; disease burden assessed by flow cytometry to measure percent GFP+ blasts (****=$p \leq 0.0001$, two-tailed unpaired t-test, from left to right: n = 6, 7, 7, 7, Means ± SD are shown). **g** A representative example of the data summarized in Fig. 2f (Gating strategy for sorting GFP Positive cells was live cells>cell profile>single cells>GFP positive cells.). **h** A survival analysis of mice transplanted with Trib2-driven AML according to treatment condition (****=$p \leq 0.0001$, from Kaplan–Meier estimator). **i** Primary murine MLL-AF9 AML blasts treated in vitro with vehicle, 200 nM CPI203, 2.5 μM nutlin-3, and the drug combination for 72 h. Cell viability assessed by resazurin (Means ± SD are shown, $n = 3$). **j** Analysis of disease burden in peripheral blood, based on percent CD11b$^{low}$ Gr1+ blasts, after 7 days of treatment, according to drug condition (***=$p \leq 0.001$,****=$p \leq 0.0001$, two-tailed t-test, from left to right, n = 4, 9, 9, 10, Means ± SD are shown). **k** A survival analysis of mice transplanted with MLL-AF9-driven AML, according to treatment condition (****=$p \leq 0.0001$, from Kaplan–Meier estimator).

To confirm that these findings were not confined to this leukemia, given the heterogeneous nature of AML, we sought to demonstrate that the drug combination could also elicit improved killing in a distinct AML mouse model, namely the MLL-AF9 mouse model. This aggressive AML, driven by a *KMT2A* translocation occurring in ~5% of human adult AML, is well-established and frequently utilized for testing novel drugs for AML[32]. We first confirmed by DNA sequencing that murine MLL-AF9 cells harbored wild-type p53 (data not shown), and that at most drug ratios the drug combination enhanced killing of MLL-AF9 myeloblasts in vitro (Fig. 2i). Sub-lethally irradiated cohorts of mice were injected with GFP+ MLL-AF9 AML blasts and cohorts of comparable disease burden (Supplementary Fig. 2F) were treated with vehicle, single drugs or combination for 21 days. Analysis of disease burden in peripheral blood after 7 days of treatment, based on GFP+ blasts and CD11b$^{low}$ Gr1-expressing immature blasts known to expand in AML[32], demonstrated the superiority of the drug combination over single agents in disease suppression (CDI = 0.36, Fig. 2j and Supplementary Fig. 2G). Moreover, although at this high dose of leukemic cells (200,000 per mouse) the vehicle-treated mice succumbed to disease in less than 2 weeks, the drug combination-treated mice showed a robust and highly statistically significant survival advantage compared with the single agent-treated mice in the MLL-AF9 model (Fig. 2k). As expected, the 4 treatment cohorts showed comparable disease burden at the time of cull based on similar disease symptoms (Supplementary Fig. 2H). We conclude that in two different p53 wild-type mouse models of AML, the drug combination is superior to the single agents in suppressing disease and extending survival.

**BETi potentiate activation of p53 by MDM2i.** We set out to define the basis of the enhanced combined toxicity of BETi and MDM2i towards AML cells. In the OCI-AML3 cell line, treatment with single-agent CPI203 reduced the protein abundance of c-MYC as expected, but this was not substantially further reduced by the drug combination (Fig. 3a). Likewise, nutlin-3 modestly increased abundance of p53, but again this not enhanced by the drug combination (Fig. 3a). Therefore, synergy does not appear to result from a concerted effect of the drugs on stabilization of p53 nor repression of c-MYC. Previously, a combination of MDM2i and the BCL2 inhibitor Navitoclax has been shown to synergize in killing AML cells[10]. Since BETi downregulate expression of BCL2 and upregulate expression of BCL2 inhibitor BIM[14,33], we wondered whether BETi-mediated downregulation

of BCL2 activity contributes to synergy between BETi and MDM2i. We confirmed that BETi CPI203 downregulates BCL2 (Fig. 3b). However, ectopic expression of BCL2 did not suppress killing by the combination (Fig. 3c, d). These results do not eliminate a role for regulation of BCL2 family proteins (including BIM) in drug combination-induced AML cell killing, but do indicate that BETi-mediated repression of BCL2 is not necessary for killing by the combination.

To obtain unbiased insight into the potential basis of enhanced combined toxicity, we set out to compare the gene expression profiles of OCI-AML3 cells treated with either vehicle, CPI203 (200 nM), nutlin-3 (2.5 μM), or the drug combination. OCI-AML3 cells were treated with vehicle, CPI203 200 nM, nutlin-3 2.5 μM, and the drug combination for 24 h, then RNA was extracted and analyzed by RNA-seq. Principal Component Analysis confirmed the expected difference between the drug treatments at the gene expression level (Supplementary Fig. 3A). Following treatment with CPI203, either as a single agent or as part of the drug combination, ~6,100 coding genes significantly changed expression relative to vehicle (Supplementary Fig. 3B–D). A much smaller number of genes (174) significantly changed expression following treatment with nutlin-3 only (Supplementary Fig. 3B–D). This was anticipated given the broad effects of BET inhibition, at least in part a consequence of down-regulation of c-MYC, compared with a more restricted number of known p53 target genes[14–16]. We reasoned that synergistic killing by the drug combination might be underpinned by synergistic changes in gene expression. Indeed, visual analysis of a heatmap of significantly changed genes revealed clusters of apparently synergistically up and down-regulated genes (Supplementary Fig. 3E). Quantitative analysis yielded 252 genes that were synergistically up-regulated and 94 genes that were synergistically down-regulated by the drug combination (Fig. 3e and Supplementary Data 1 and 2). Ingenuity Pathway Analysis (IPA) analysis showed that the synergistically down-regulated genes were enriched in cell cycle genes (Supplementary Fig. 3F). The synergistically up-regulated genes were most significantly enriched in genes involved in the p53 pathway (Supplementary Fig. 3 F). Of 116 known high-confidence p53 target genes[34], 24 were synergistically upregulated by the drug combination (Fig. 3f and Supplementary Fig. 3G). Synergistically up-regulated *TP53* target genes included *ANKRA2, ARHGEF3, BBC3 (PUMA), BTG2, CDKN1A, DRAM1, FUCA1, GADD45A, GDF15, IER5, LAPTM5, MDM2, PMAIP1, RAP2B, RRM2B, SERTAD1, SESN2, TGFA, TNFRSF10B, TNFRSF10D, TP53I3, TP53INP1, ZMAT3* and *ZNF337* (Fig. 3f). Representative sequence tracks of p53 target genes *GDF15, CDKN1A* and *BBC3* are shown (Supplementary

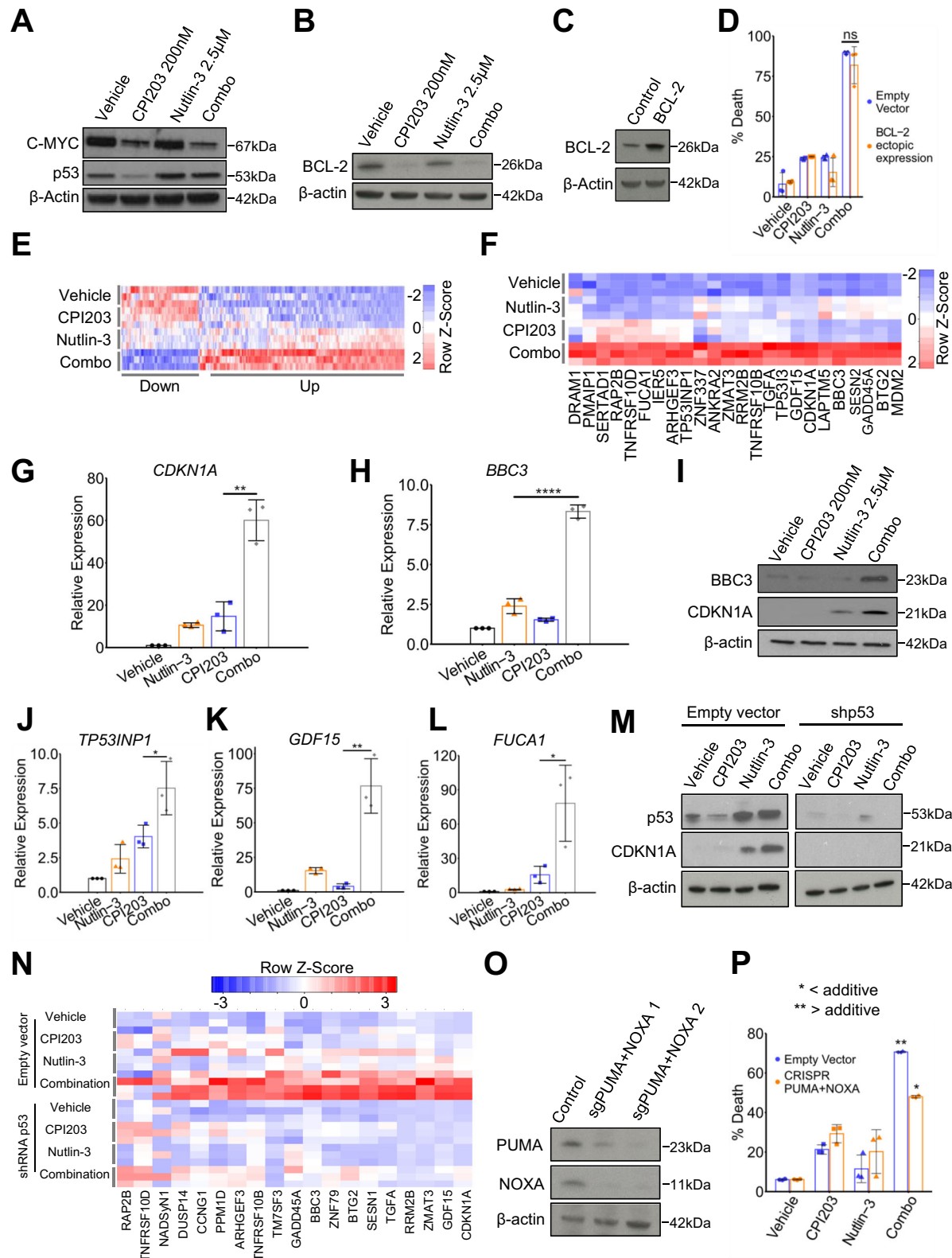

Fig. 3H). Synergistic upregulation of p53 targets, *CDKN1A* and *BBC3*, by the drug combination was confirmed at the RNA level by qPCR and the protein level by western blot (Fig. 3g–i), and at the RNA level for *GDF15, TP53INP1* and *FUCA1* (Fig. 3j–l). We conclude that enhanced toxicity of the drug combination is associated with enhanced expression of p53 target genes.

To confirm that expression of these p53 target genes is indeed dependent on p53, we generated cells expressing shRNA to knock down p53 (or empty vector control cells) and treated with single or combination drugs (Fig. 3m). p53-deficient cells failed to upregulate CDKN1A in response to nutlin-3 or combination (Fig. 3m and Supplementary Fig. 3I). Principal Component

**Fig. 3 BET inhibitors potentiate activation of p53 target genes by p53. a** Western blot of C-MYC and p53 in OCI-AML3 cells treated for 24 h with vehicle, 200 nM CPI203, 2.5 μM nutlin-3, and the drug combination. **b** Western blot for BCL-2 in OCI-AML3 cells treated for 24 h with vehicle, 200 nM CPI203, 2.5 μM nutlin-3, and the drug combination. **c** Western blot for BCL-2 comparing the control OCI-AML3 and cells over-expressing BCL-2. **d** Cell killing as assessed by flow cytometry using Annexin-V and PI staining of OCI-AML3 control (empty vector) cells versus OCI-AML3 over-expressing BCL-2, according to treatment condition (p-value >0.05 by two-tailed unpaired t-test, Means ± SD are shown, n = 3). **e** A heat map of genes synergistically regulated by the drug combination determined by RNA-seq of OCI-AML3 cells according to treatment condition (24 h). Genes synergistically up-regulated (Up) by the drug combination were rigorously identified as those where C/(A + B) = > 1.25 and combination FPKM > DMSO FPKM, and synergistically down-regulated genes (Down) where C/(A + B) = > 1.25 and combination FPKM < DMSO FPKM (where C = combination FPKM – DMSO FPKM; A = nutlin-3 FPKM – DMSO FPKM; B = CPI203 FPKM – DMSO FPKM). **f** A heat map of 24 high confidence p53 target genes that are synergistically up-regulated or down-regulated in OCI-AML3 according to treatment condition. **g** qPCR analysis of *CDKN1A* expression in OCI-AML3 under indicated treatments for 24 h (**=p ≤ 0.01, two-tailed unpaired t-test, Means ± SD are shown, n = 3). **h** qPCR analysis of *BBC3* expression in OCI-AML3 under indicated treatments for 24 h (****=p ≤ 0.0001, two-tailed unpaired t-test, Means ± SD are shown, n = 3). **i** Western blot analysis of p53 targets BBC3 and CDKN1A in OCI-AML3 according to treatment condition. **j** qPCR expression analysis of *TP53INP1* in OCI-AML3 under indicated treatments for 24 h (*=p ≤ 0.05, two-tailed unpaired t-test, Means ± SD are shown, n = 3). **k** qPCR expression analysis of *GDF15* in OCI-AML3 under indicated treatments for 24 h (**=p ≤ 0.01, two-tailed unpaired t-test, Means ± SD are shown, n = 3). **l** qPCR expression analysis of *FUCA1* in OCI-AML3 under indicated treatments for 24h (*=p ≤ 0.05, two-tailed unpaired t-test, Means ± SD are shown, n = 3). **m** Western blot analysis of p53 and CDKN1A in OCI-AML3 cells and OCI-AML3 cells harboring shRNA p53 according to treatment condition. **n** A heat map of an RNA-seq data of 19 high confidence p53 target genes that are synergistically up-regulated by the drug combination in control (empty-vector) OCI-AML3 cells, versus shRNA p53 OCI-AML3 cells, according to treatment condition. **o** Western blot analysis of BBC3 and NOXA in control (empty vector) OCI-AML3 cells, and OCI-AML3 cells harboring CAS9 and sgRNAs against PUMA and NOXA. **p** FACS analysis of cell death (annexin VI and PI) in control OCI-AML3 cells or cells in which PUMA and NOXA were knocked out by CRISPR/CAS9, under indicated conditions for 72 h (*=<additive. **=>additive, two-tailed unpaired t-test, Means ± SD are shown, n = 3).

Analysis (PCA) of RNA-seq data confirmed that p53 knock down blunted the effect of the drug combination on the whole transcriptome (data not shown). Similar to the previous experiment, 214 genes were calculated to be synergistically upregulated by the drug combination in control cells. Of these, 204 were dependent on p53 for their synergistic upregulation (Supplementary Data 3). Of the 116 *bona fide* p53 target genes[34], 19 were upregulated in control AML3 cells expressing p53 (12 of which were in common with the previous experiment (Fig. 3f, n and Supplementary Fig. 3J). None of these 19 p53 target genes was synergistically upregulated in drug combination-treated p53-deficient cells (Fig. 3n). To dissect which p53 target genes are required for enhanced combined drug toxicity in OCI-AML3 cells, we used CRISPR/Cas9 to generate derivatives of OCI-AML3 cells lacking *CDKN1A*, *BBC3* and *NOXA*. Consistent with the observation that drug combination synergy is linked to cell death (Fig. 1), inactivation of CDKN1A, a well-known effector of p53-mediated cell cycle arrest but not apoptosis, did not affect synergy of the combination (Supplementary Fig. 3K). More surprisingly, inactivation of pro-apoptotic p53-target genes *BBC3* and *NOXA* on their own did not significantly affect cell killing (data not shown). However, combined inactivation of *BBC3* and *NOXA* markedly blunted drug combination-induced cell killing (Fig. 3o, p). Together, these results unexpectedly show that BET inhibition potentiates activation of p53. WT p53 is required for enhanced expression of p53 target genes by BET inhibition and at least two of these genes, *BBC3* and *NOXA*, are required for enhanced combined toxicity towards AML by dual BET and MDM2 inhibition.

**BET inhibition relieves BRD4-mediated repression of p53 target genes.** We set out to define the molecular mechanism by which BET inhibition potentiates activation of p53. Induction of BBC3 by nutlin-3 was also potentiated by BRD4 knock down, confirming that BETi acts, at least in part, by inhibition of BRD4 (Fig. 4a, b). We initially considered three non-mutually exclusive testable possibilities for how BETi/inhibition of BRD4 potentiates activation of p53 by MDM2i. First, we postulated that the BETi might stabilize p53 target mRNAs. Second, we considered the possibility that BETi might, directly or indirectly, promote expression of known p53 activators that synergize with p53 stabilized by MDM2i to potentiate activation of the p53

pathway. Third, we hypothesized that BETi might promote p53 binding to its cognate target genes. To test whether BETi stabilized expression of p53 target mRNAs, OCI-AML3 cells were treated with vehicle, single drugs or combination and then 2 h later with actinomycin D to inhibit new transcription. Abundance of p53 target mRNAs was determined by qPCR over a time-course, allowing relative assessment of mRNA half-life. Figure 4c, d demonstrates that, compared to either single agent, the drug combination did not promote stability of *CDKN1A* or *BBC3* mRNAs.

To test the role of selected p53 activators, we mined our RNA-seq data for known p53 activators upregulated by BETi. Based on analysis of RNA-seq data, at least two known p53 activators, *JNK* and *53BP1*[35,36], were upregulated by BETi alone and the drug combination (Fig. 4e). qPCR confirmed significant upregulation of *JNK* by BETi and an upward trend for *53BP1* (Fig. 4f, g). To test a role for JNK or 53BP1 in activation of p53, we knocked them down by lentivirus-delivered shRNA (Fig. 4h, i). However, neither was required for enhanced toxicity of the MDM2i and BETi combination (Fig. 4j, k).

To test whether the drug combination promoted binding of p53 to its target genes compared to either single agent alone, we performed ChIP-seq to assess p53 binding across the whole genome in OCI-AML3 cells. Three independent replicates of OCI-AML3 cells were treated with vehicle control, CPI203 200 nM, nutlin-3 2.5 μM or the drug combination. Cells were harvested for ChIP 6 h after drug treatment, since qPCR and western blot analysis showed that upregulation of p53 target genes was already detectable at this time (data not shown). Analysis of ChIP-seq data showed that the mean p53 peak width under all treatment conditions was ~400 bp (Supplementary Fig. 4A). The overall genome distribution of sites under all conditions was similar and very few sites were unique to drug combination-treated cells (Supplementary Fig. 4B, C). In fact, the drug combination tended to decrease the total number of p53 binding sites, compared to nutlin-3 alone (Fig. 4l, Supplementary Fig. 4C, D). Most of the 252 synergy up genes did not bind p53 in either nutlin-3 or combination-treated cells (Supplementary Fig. 4E), suggesting that they are not direct p53 targets and their activation by nutlin-3 is indirect. For the 24 synergy up genes that are also *bona fide* p53 target genes, most bound p53 in the nutlin-3-treated cells, as expected, but this was not increased in

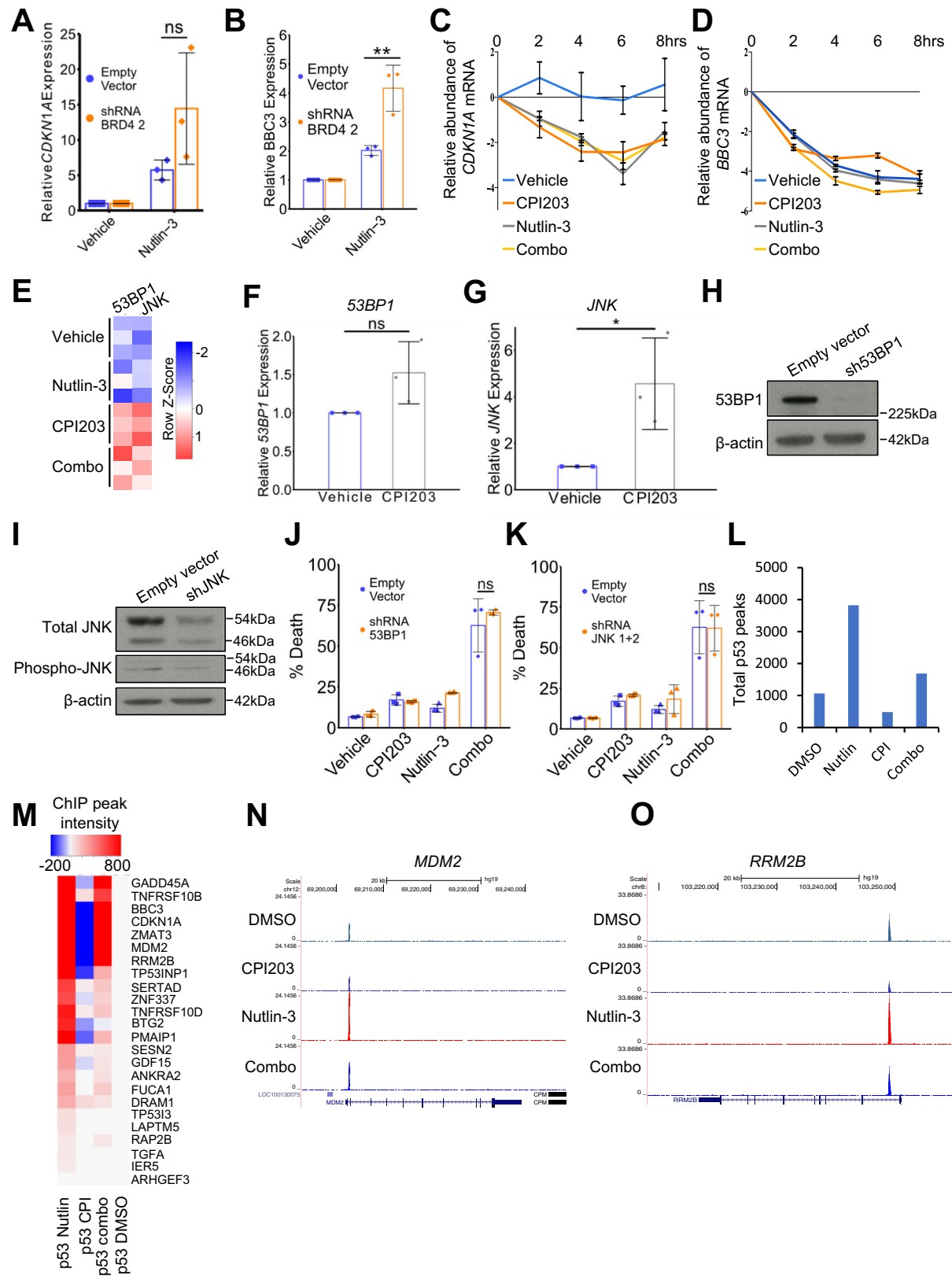

combination-treated cells (Fig. 4m). Instead, at p53 target genes, p53 binding tended to decrease in combination-treated cells compared to nutlin-3-treated cells (Fig. 4m), in line with the global analysis (Fig. 4l and Supplementary Fig. 4C, D). Examination of the sequence tracks for individual genes, such as *MDM2* and *RRM2B*, confirmed the observation that BETi, in combination with nutlin-3, did not increase the binding of p53 to its target genes (Fig. 4n, o). We ultimately conclude that BET inhibition does not activate p53 by stabilizing its target gene mRNAs, nor via promotion of activity of selected tested known p53 activators, nor by increasing p53's binding to its target genes.

**Fig. 4 BET inhibitors do not stabilize p53 target mRNAs nor increase binding of p53 to target genes. a** qPCR assessment of expression of *CDKN1A* in control (empty-vector) OCI-AML3 cells versus OCI-AML3 cells harboring shRNA against *BRD4*, comparing treatment with vehicle and 2.5 μM nutlin-3 (*p*-value >0.05 by two-tailed unpaired t-test, Means ± SD are shown, *n* = 3). **b** qPCR assessment of expression of *BBC3* in control (empty-vector) OCI-AML3 cells versus OCI-AML3 cells harboring shRNA against *BRD4*, comparing treatment with vehicle and 2.5 μM nutlin-3 (two-tailed unpaired t-test, Means ± SD are shown, *n* = 3, ** means p-value < 0.01). **c** qPCR analysis of expression of *CDKN1A* in OCI-AML3 cells treated with Actinomycin D over the indicated time-course (0, 2, 4, 6, 8 h) and under the indicated drug treatments (Means ± SD are shown, *n* = 3). **d** qPCR analysis of *BBC3* in OCI-AML3s treated with Actinomycin D over the indicated time-course (0, 2, 4, 6, 8 h) and under the indicated drug treatments (Means ± SD are shown, *n* = 3). **e** Heat map showing expression of *JNK* and *53BP1* in OCI-AML3 cells after the indicated drug treatments for 24 h. **f** qPCR expression analysis of *53BP1* in OCI-AML3 cells after indicated drug treatments for 24 h (two-tailed unpaired t-test, Means ± SD are shown, *n* = 3). **g** qPCR expression analysis of *JNK* in OCI-AML3 cells after indicated drug treatments for 24 h (*=p ≤ 0.05, two tailed unpaired t-test, Means ± SD are shown, *n* = 3). **h** Western blot of 53BP1 in OCI-AML3 expressing either empty vector or shRNA *53BP1*. **i** Western blot of *JNK* in OCI-AML3 cells expressing either empty vector or shRNA *JNK* isoform 1 + isoform 2. **j** FACS analysis of cell death (annexin VI and PI) in OCI-AML3 cells expressing empty-vector or shRNA *53BP1* after indicated drug treatments for 72 h. **k** FACS analysis of apoptosis (annexin VI and PI) in OCI-AML3 cells expressing empty-vector or shRNA *JNK* isoform 1 + isoform 2 after indicated drug treatments for 72 h. **l** Total number of p53 binding sites across the genome (ChIP-seq) after indicated drug treatments for 6 h, present in at least 2 out of 3 replicates. **m** Heat map of p53 binding at indicated genes in OCI-AML3 cells after indicated drug treatments for 6 h. **n** Sequence tracks of p53 binding at *MDM2* gene after indicated drug treatments for 6 h. **o** Sequence tracks of p53 binding at *RRM2B* gene after indicated drug treatments for 6 h.

At this point, closer inspection of the RNA-seq data revealed that p53 was ranked as one of the top upstream regulators of genes differentially expressed by CPI203 (Supplementary Fig. 5A). Focused analysis of p53 target genes directly showed that CPI203 alone was sufficient to increase expression of many p53 target genes in OCI-AML3 cells (Supplementary Fig. 5B, C). For several p53 target genes this was confirmed by qPCR after treatment with CPI203 or knock down of BRD4 by lentivirus-encoded shRNA (Supplementary Fig. 5d, e). Inhibition of BET family proteins with a potent PROTAC BET degrader BETd-260 strongly upregulated expression of p53 target gene p21 at the protein level (Supplementary Figs. 1N and 5F)[28]. Since previous results established a critical role for BRD4 (Figs. 1n, o and 4a, b), these data raised the possibility that BRD4 might be a repressor of p53 target gene expression. To test this, we better characterized the effect of BETi and nutlin-3 on BRD4's chromatin binding. We performed ChIP-seq analysis to determine genomic localization of BRD4 in control, single drug and combination-treated cells. In control OCI-AML3 cells, BRD4 reproducibly bound ~9800 sites occupying ~27 Mb across the genome (Fig. 5a, b and Supplementary Fig. 6A). In nutlin-3-treated cells, a similar number of BRD4 binding sites and total genome occupancy was observed (Fig. 5a, b and Supplementary Fig. 6B), and the majority of binding sites overlapped between control and nutlin-3-treated cells (Supplementary Fig. 6C). As expected, in BETi and combination-treated cells, the number of reproducible BRD4 binding sites was greatly decreased (Fig. 5a, b and Supplementary Fig. 6C–E). In both control and nutlin-3-treated cells the majority of the BRD4 binding sites were at annotated gene promoters (Fig. 5C). Across all genes, in vehicle-treated cells BRD4 binding correlated with gene expression (Supplementary Fig. 6F) and globally a loss of BRD4 on treatment with BETi was associated with a small but significant decrease in gene expression (Supplementary Fig. 6G), consistent with the role of BRD4 as a global transcriptional activator[12].

To evaluate the role of BRD4 in potentiation of p53 target gene expression by BETi, we considered BRD4 binding in nutlin-3 vs combination-treated cells. For the 24 synergy up genes that are also *bona fide* p53 target genes (Fig. 3f), 22 bound BRD4 in nutlin-3-treated cells but only 2 bound BRD4 in combination-treated cells (Fig. 5d). This interpretation was supported by analysis of individual gene loci, e.g. *CDKN1A* and *BBC3* (Fig. 5e, f). As noted above, the vast majority of the synergy up p53 target genes also bind p53 in both nutlin-3 and combination-treated cells (Figs. 4n, n and 5d). In other words, synergy up p53 target genes, tend to bind p53 and BRD4 in nutlin-3-treated cells but p53 only in combination-treated cells (Fig. 5d–f). These results

suggest that in AML cells, BRD4 may act as a repressor of p53 target genes. In line with this idea, treatment of OCI-AML3 cells with CPI203 modestly activated a number of p53 target genes, even in the absence of nutlin-3, e.g., *PMAIP1, RAP2B, FUCA1,* and others (Fig. 3f). To directly test the ability of BRD4 to repress p53 target genes, we ectopically expressed the short form of BRD4 (BRD4S) in AML3 cells treated with nutlin-3 to stabilize p53. Cells ectopically expressing BRD4S activated p53 target genes *CDKN1A* and *PUMA* to a lesser extent than empty vector cells upon treatment with nutlin-3 (Fig. 5g–i). Measured against a number of other p53 target genes, there was an invariable trend towards repression by BRD4S (Supplementary Fig. 7A–H). The repressive effect of BRD4S was confirmed using a stably integrated p53 fluorescent/luminescent reporter gene (Supplementary Fig. 6I, J). BRD4S also repressed p53 target gene expression in wild-type *TP53* MOLM13 cells (Supplementary Fig. 7K–N). These results suggest that in AML, BRD4 is able to bind p53 target genes and can repress their activation, even when p53 is bound. Displacement of BRD4 by BET inhibition relieves this repression, leaving p53 free to activate its pro-apoptotic targets, thereby accounting for the enhanced killing of AML by combined MDM2 and BET inhibition.

## Discussion

In this study, we show that BRD4 inhibition together with p53 activation results in enhanced anti-leukemia activity both in vitro and in vivo, in AML retaining wild-type *TP53*, compared to either inhibition alone. Apparent synergy depends on wild-type *TP53*, is linked to enhanced activation of p53 by BETi and depends on the expression of at least some p53 target genes, namely *BBC3* and *NOXA*. Regarding the mechanism underlying this enhanced anti-leukemia activity, we eliminated some obvious candidates, such as the binding of p53 to its target genes and the stabilization of p53 target mRNAs. Instead, we present evidence that the basis of this enhanced anti-leukemia activity is via relief of an unexpected repressive effect of BRD4 on expression of p53 target genes, thereby unleashing the full pro-apoptotic activity of p53 stabilized by MDM2i.

Through its role as a "reader" of acetylated lysine residues, BRD4 is widely regarded as a driver of transcriptional activation. Mechanistically, BRD4 promotes gene expression via the recruitment and activation of P-TEFb, which drives RNA polymerase II-dependent transcription[37]. However, some reports have implicated BRD4 in transcriptional repression. Although some reports have implicated BRD4S preferentially as the repressive isoform, for example of transcriptionally silent latent HIV virus by recruitment of repressive SWI/SNF chromatin remodeling

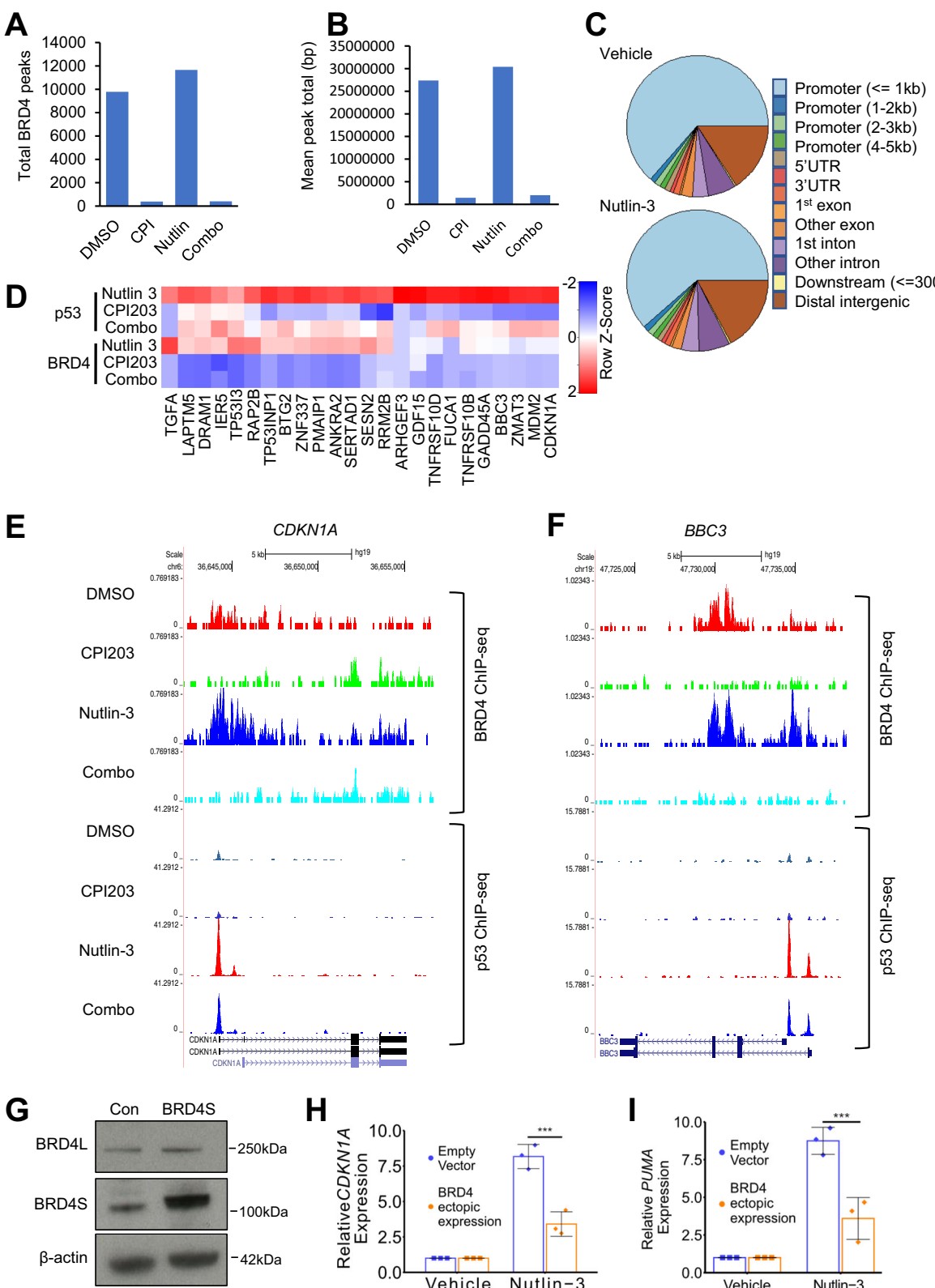

complexes[38], other studies have shown that BRD4L also acts as a repressor, for example of the HPV-encoded E6 gene and autophagy genes by recruitment of histone methyltransferase G9A[39,40]. Conceivably, a BRD4/G9A interaction is also involved in the repression of p53 target genes. Previous studies have demonstrated a physical interaction between p53 and BRD4[26,41]. Wu et al showed that BRD4 promotes the expression of p53 target

gene, *CDKN1A*, at least in 293 cells[41]. These studies raise the possibility that in some contexts a p53/BRD4 physical interaction promotes expression of p53 target genes, but in AML cells BRD4's interaction with p53 is co-opted into a repressive interaction that silences p53 target genes and facilitates leukemogenesis. Our demonstration that BRD4 acts as a repressor and silences p53 activity extends the repressive activity of BRD4 into an important

**Fig. 5 BRD4 represses p53 target genes. a** Total BRD4 peaks determined by ChIP-seq (present in both replicates) in OCI-AML3 cells after indicated drug treatments (6 h). **b** Mean total base pairs covered by BRD4 ChIP-seq peaks (peaks present in 2 out of 2 replicates) after the indicated drug treatments of OCI-AML3 cells. **c** Genomic distribution of BRD4 binding sites identified by ChIP-seq (peaks present in 2 out of 2 replicates), after the indicated drug treatments of OCI-AML3 cells. **d** Heat map showing BRD4 ChIP-seq (peaks present in 2 out of 2 replicates) at 24 p53 target genes, after the indicated drug treatments of OCI-AML3 cells. **e** Sequence tracks of representative BRD4 and p53 binding at *CDKN1A*, after the indicated drug treatments of OCI-AML3 cells. **f** Sequence tracks of representative BRD4 and p53 binding at *BBC3*, after the indicated drug treatments of OCI-AML3 cells. **g** Western blot for BRD4 in OCI-AML3 cells ectopically expressing the short isoform of BRD4 (BRD4S). BRD4L is the long isoform. **h** qPCR analysis of *CDKN1A* in OCI-AML3 cells ectopically expressing BRD4S, in absence or presence of nutlin-3 (***=p ≤ 0.001, two-tailed unpaired t-test, Means ± SD are shown, n = 3). **i** qPCR analysis of *BBC3A* in OCI-AML3 cells ectopically expressing BRD4S, in absence or presence of nutlin-3 (***=p ≤ 0.001, two-tailed unpaired t-test, Means ± SD are shown, n = 3).

new context and has important implications for AML pathogenesis and candidate therapeutic approaches.

Regarding pathogenesis, across all human cancers, the *TP53* tumor suppressor gene is the most frequently mutated gene[42]. In mice, inactivation of p53 cooperates with activated RAS in leukemogenesis[43–45]. However, although AML is, overall, a heterogeneous disease, surprisingly more than 90% of de novo AML retain wild-type *TP53*[2,3], suggesting that human AML subtypes employ alternative mechanisms to inactivate the p53 pathway[5]. At least some AML dysregulate known p53 regulators, MDM2, MDM4, and ARF[5]. Our data suggest that in some AML dysregulation of BRD4 might also antagonize the p53 pathway to facilitate leukemogenesis. Consistent with this idea, *BRD4* exhibits elevated expression in ~7% of AML[46], and, in the majority of cases, this is accompanied by wild-type *TP53* (cBioportal).

With respect to novel candidate therapies for AML, our studies suggest promising efficacy of the MDM2i and BETi combination in AML. Previous studies have suggested that the efficacy of an MDM2i and BETi combination in chronic myeloid leukemia (CML) comes from dual targeting of the p53 and c-MYC pathways, by MDM2i and BETi respectively[47]. As a secondary consequence of BETi-mediated downregulation of c-MYC, BETi can activate at least one p53 target gene, *CDKN1A*, through loss of c-MYC-mediated gene repression[48]. Thus, although our studies indicate that *TP53* wild type AML are typically sensitive to the MDM2i/BETi combination, the dominant mechanisms underlying synergistic toxicity are likely to vary, especially given the genetic heterogeneity of AML[2,3]. At least in OCI-AML3 cells (representative of a recurrent AML genotype found in 10-15% of AML, *NPM1*, and *DNMT3A* mutant and *TP53* wild type), we have shown an unexpected ability of BETi to directly potentiate activation of p53 by MDM2i, via relief of BRD4-mediated gene repression. Also in two mouse models, we observed markedly enhanced anti-AML activity of the drug combination, compared to either single drug alone. Of note here, however, we used a clinical-grade MDM2i, RG7112, that has been optimized against human MDM2i. Therefore, it is possible that our mouse studies diminish the on-target toxicity of MDM2i toward normal healthy mouse tissues, thereby increasing the tolerability and apparent therapeutic window of the drug in this experimental setting. Further validation of this novel drug combination, in terms of efficacy and tolerability, in both animal models and human cells is warranted.

The molecular heterogeneity of AML has been a hurdle to the development of novel therapies of benefit to a substantial proportion of patients. For example, *NPM1*, the most commonly mutated gene in AML, is mutated in only 28% of AML and many genes are recurrently mutated, but in less than 10% of patients, for example, *ASXL1*, *IDH2*, *RUNX1,* and *SRSF2*[2,3]. In contrast, in pre-clinical studies, a substantial proportion of AMLs are relatively sensitive to BETi and ~90% of AML express wild-type *TP53*[2,3,14–16,48] suggesting that the MDM2i and BETi drug combination can potentially target a majority of AML.

In summary, we demonstrate that MDM2 and BET inhibition are synthetically lethal in AML with wild-type p53. We propose that BRD4 represses transcription of p53 target genes, such that when BETi block repression in the context of p53 stabilization by MDM2i the p53 pathway is potently activated leading to enhanced anti-leukemia activity, compared to either drug alone. As single agents these compounds have been shown to be effective and quite well-tolerated in clinical trials[49]. Given the superiority of the combination over single drugs in our in vitro and in vivo studies, a clinical trial employing these two agents as a combination in AML retaining wild-type TP53 is justified.

## Methods

**Primary AML cells**. A total of 15 AML patient samples were studied. All primary bone marrow aspirates were taken from routine diagnostic specimens after the informed consent of the patients. The project received approval from the local ethics committee (Brighton and Sussex University Hospitals NHS Trust Research and Development Committee) as The Brighton Blood Disorder Study, references 09/025/CHE and 09/H1107/1) and was conducted in accordance with the Declaration of Helsinki. Mononuclear cells from patients diagnosed with AML were isolated by Histopaque 1077 density gradient purification. They were stored in the Brighton and Sussex Medical School (BSMS) tissue bank and plated at a density of 40,000 cells/per well in the black 96 wells plates. Cells were plated in 80 μl of RPMI containing 10% FCS, 100 mM glutamine, 10,000 I.U/mL Penicillin, and Streptomycin. Relevant co-variate data on patients can be found in Supplemental Data 8.

**AML Cell lines**. THP1 was obtained from ATCC, KASUMI-1, MOLM-13, and OCI-AML3 from DSMZ, KG1a, and MV411 were gifted from Professor Mhairi Copland and Dr. Xu Huang, respectively (both Paul O'Gorman Leukemia Research Centre, Glasgow). The authenticity of all cell lines was confirmed by genotyping. Cell lines were grown according to the vendors' instructions, were incubated at 37 °C in a humidified incubator with 5% CO₂, and passaged every 3–4 days. For p53 activity assays with luciferase and GFP reporters, AML3 cells were stably infected with pGF-p53-mCMV-EF1α-Puro lentivirus (https://systembio.com/shop/pgf-p53-mcmv-ef1α-puro-ht1080-stable-cell-line/).

**Mice**. Study was approved by University of Glasgow Animal Welfare & Ethical Review Board (AWERB) and carried out under UK Home Office regulation. Trib2-expressing AML blasts[30] from serial transplants were thawed and maintained in medium (DMEM with 15% FBS + 100 units/ml penicillin-streptomycin and 2 mM L-glutamine) and supplemented with 10 ng/ml IL-3 (Peprotech, 213-13), 10 ng/ml IL-6 (Peprotech 216-1) and 100 ng/ml SCF (Peprotech 250-03). MLL-AF9 cells[32] were maintained in medium (RPMI with 20% FBS + 100 units/ml penicillin streptomycin and 2mM L-glutamine) and supplemented with 10 ng/ml IL-3 (Peprotech, 213-13). For in vivo experiments, mice were sub-lethally irradiated at 5.5 Gy, and 4 h later injected with 850,000 cells for the Trib2 murine AML model and 200,000 leukemia cells for the MLL-AF9 murine experiment. For both experiments a final volume of 200 μl of leukemia cells was injected through the tail vein. Mice were maintained on Baytril antibiotic in their drinking water pre- and post-transplantation, for two weeks. Mice were bled via tail vein weekly to test for disease engraftment; when this was confirmed drug treatment was initiated. For in vivo studies, CPI0610 (Constellation Pharmaceuticals) was dissolved in heated 0.5% methylcellulose then sonicated (using a Diagneode Bioruptor). RG7112 was mixed with its vehicle supplied by Roche (2% hydroxypropyl cellulose, 0.1% polysorbate 80, 0.09% methyl paraben, 0.01% propyl paraben). For the Trib2 fixed endpoint experiment, Trib2 mice were treated for 21 days. RG7112 was initially given once daily 100 mg/kg and CPI0610 twice daily 30 mg/kg, both by oral gavage. However, due to excessive weight loss of one of the combination-treated mice, the mice received a 2 day "drug holiday" after 10 days and resumed dosing at once daily RG7112 70 mg/kg and twice daily CPI0610 30 mg/kg. For the Trib2 and

MLL-AF9 survival experiments, mice initiated 21 days of drug dosing (not including 2 drug-free days every 5 days) (RG7112 was given once daily 70 mg/kg and CPI0610 was given twice daily 30 mg/kg, both by oral gavage). After reaching the end of treatment, mice were culled immediately or maintained to an ethical survival endpoint.

**Drug treatments of cells in vitro**. For drug treatment of primary AML blasts, 2.5 μM nutlin-3 alone, 200 nM CPI203 alone, and the two in combination were added to triplicate wells to a volume of 100 ul. Control experiments were performed without the addition of drugs using vehicle (DMSO) only. After 48 h cell viability was measured using the CellTiter-Glo reagent (Promega, G7572), and the luminescence was detected with Biotek synergy HT plate reader and analyzed using Gen 5 version 1.08 software. Cell viability was also determined by trypan blue dye exclusion in some experiments (Sigma, T8154). Apoptosis was assessed using Annexin V and propidium iodide (PI) staining (BD Biosciences; 556547) as per manufacturer's instructions and analyzed using the BD Fortessa flow cytometer. For p53-promoter based dual GFP/Luminescence assay after 24 h 2.5 μM nultin-3 treatment, cells were lysed using 1xpassive buffer (Promega, E1941) and lysate was either measured directly for GFP signal or incubated with luciferase assay reagent LAR (Promega, E1500) for luminescence signal. The GFP and luminescence were detected with CLARIOstar plate reader and analyzed using Microsoft Excel.

Drug combination and synergy analyses

Chou-Talalay Combination Index method: AML cell lines were treated with combinations of CPI203 (Constellation Pharmaceuticals) and nutlin-3 (Sigma-Aldrich, N6287) (CPI203 ranging from 12.5 nM to 800 nM and nutlin-3 ranging from 1.25 μM to 10 μM) to determine synergistic doses. Cell viability was determined after 72 h by either trypan blue dye (Sigma, T8154) exclusion or resazurin (Alamar blue dye, Sigma) with the Envision Fluorescent Reader (Perkin Elmer). Mean fluorescent values from multiple replicates were calculated for each condition. Assessment of synergy was made by calculating combination indices (CI) using Calcusyn software (version 2.0), CI < 1 considered synergistic, CI = 1 considered additive and CI > 1 considered antagonistic[25]. Bliss Independence method: AML cell lines were seeded on 384-well plates at 3,000 cells per well and treated for 72 h (or 24 h for BET PROTAC experiments) with the indicated doses (2x serial dilution) of drugs, alone or in combination. Luminescence from quadruplicate was measured using the CellTiter-Glo reagent (Promega, G7572). Bliss synergy was calculated using SynergyFinder v2.0 web-based application with default parameters for calculating bliss independence. The difference between the observed combined effect and the expected combined effect of the two drugs is called the Excess over Bliss (*eob*). Positive *eob* values are indicative of synergistic interaction, negative *eob* values are indicative of antagonistic behavior and null *eob* values indicate no drug interaction.

**RNA extraction, cDNA synthesis, and qPCR**. RNA was extracted using the RNeasy Mini kit (Qiagen, cat no. 7410) according to the manufacturer's protocol. On-column DNA digestion was carried out using DNase1 (Qiagen, cat no. 79254). Eluted RNA was quantified using the Nanodrop 2000 (ThermoFisher Scientific) and 1 μg RNA was used in cDNA synthesis (Invitrogen, 18080-093), according to the manufacturer's instructions. For cDNA synthesis: 1 μg RNA, 1 μl of 50 μM oligo DT primer (Invitrogen 18418020), 1 μl dNTP mix (10 mM each of dATP, dGTP, dCTP, and dTTP) and water to a final volume of 14 μl were mixed and heated at 65 °C for 5 min. After a 1-min incubation on ice, the following were added: 4 μl 5X first strand buffer, 1 μl 0.1 M DTT, 1 μl Superscript™ Reverse Transcriptase (Invitrogen, 18080-093), and heated at 25 °C for 5 min. Tubes were then heated to 55 °C for 1 h and the reaction was inactivated by a 15-min incubation at 70 °C. qPCR was done using 10 μl of SYBR green master mix (2X DyNAmo HS SYBR green qPCR master mix, Thermo-Scientific, F- 410), 200 nM of each primer of interest, and water was added to a final volume of 20 μl as a master mix to 1 μl of DNA in Hard Shell PCR Plates, 96-well white, Bio-Rad, (HSP9601). The primer sequences used are listed in Supplementary Data 4. The PCR reactions were performed as follows; 95 °C for 3 min, 95 °C for 10 s, 60 °C for 20 s, 72 °C for 30 s, steps 2 to 4 repeated 39 times, 72 °C for 5 min, and 65 °C for 5 s then a gradient up to 95 °C for melt curve analysis.

**mRNA stability assays**. OCI-AML3 cells were treated with Vehicle (DMSO), 200 nM CPI203, 2.5 μM nutlin-3, or the drug combination. After a 2-h incubation, cells were treated with 5 μM Actinomycin D and harvested after 0, 2, 4, and 6 h of Actinomycin D treatment. Cell pellets were stored overnight at −80 °C before RNA was extracted and cDNA synthesized for qPCR analysis as described above. Abundance of *CDKN1A* and *BBC3* mRNA at each time point was calculated relative to time 0 using the delta CT method (Ct at time 0 minus Ct of each time point).

**RNA sequencing**. Sequencing libraries were made from poly-A RNA, as recommended by Illumina, and sequenced using either an Illumina GAIIX or a NextSeq 500 sequencer. RNA-seq paired-end reads were assessed for quality using the 'FastQC' algorithm, trimmed as appropriate using the algorithm 'trim-galore' (version 3.0), then aligned to the human genome using the splice-aware aligner TopHat2. Reference splice junctions were provided by a reference transcriptome

from the Ensembl GRCh37 (hg19) build, release version 73. The Cuffdiff tool from the Cufflinks suite was used to process aligned reads and perform maximum likelihood estimation to assess transcript abundances, before calculating the differential expression of transcripts across samples. In parallel, aligned reads for genic isoforms were collated and total read counts per gene were calculated using htseq-count version 0.5.4p3, before differential expression analysis using the linear modeling tool DESeq2 was performed. Using both differential expression methods, significantly changing expression was defined as an FDR-corrected p-value ≤ 0.005. FPKM (Fragments Per Kilobase of transcript per Million mapped reads) values were then generated. Gene ontology analysis was performed using Gene Set enrichment Analysis (GSEA), DAVID (version 6.7) and Ingenuity® Pathway Analysis (IPA) software.

**ChIP-sequencing**. ChIP-seq protocol was adapted from previously[50,51]. Antibody-bound magnetic beads (anti-rabbit or anti-mouse, as appropriate) (Dynabeads beads, M280) were incubated with 10 μg of BRD4 antibody (Abcam, 128874) or 10 μg of p53 antibody (Santa Cruz, DO1) (rabbit or mouse IgG was used as a negative control (Sigma, M7023)) for 4 h at room temperature, then added to the lysed samples. During the 4-h antibody-bead incubation, $20 \times 10^6$ OCI-AML3 cells were treated with vehicle, 200 nM CPI203, 2.5 μM nutlin-3, or the drug combination for 6 h. Following this, cells were pelleted at 200 g for 5 min and re-suspended in serum free RPMI media and cross-linked for 10 min on a rocker by adding 16% methanol-free paraformaldehyde (Alfa Aesar, 43368) (final concentration 1%). Cross-linking was quenched with 2.5 M glycine (final concentration 0.125 M) for 5 min. Cells were washed twice in cold PBS. Three lysis buffers (LB1-3) were used to produce chromatin lysates and the following protease inhibitors were added to each: 1x protease inhibitor cocktail (Sigma, P8340) and 50 μg/ml PMSF (Sigma, P7626). In total 10 ml of lysis buffer 1 (LB1) (50 mM Hepes-KOH, pH 7.5; 140 mM NaCl; 1 mM EDTA; 10% glycerol; 0.5% NP40 and 0.25% Triton X-100) was added to the pellet of cross-linked cells and this was rocked at 4 °C for 10 min, then centrifuged at 2,000 x g for 4 min at 4 °C. The supernatant was discarded and the pellet then re-suspended in 10 ml of LB2 (10 mM Tris-HCl, pH 8.0; 200 nM NaCl; 1 mM EDTA; 0.5 mM EGTA), and rocked at 4 °C for 5 min. The suspension was again pelleted, the supernatant discarded and the pellet then re-suspended in 2 ml of LB3 (10 mM Tris-HCl, pH 8; 100 mM NaCl; 1 mM EDTA; 0.5 mM EGTA; 0.1% Na-deoxycholate; 0.5% N-lauroylsarcosine). This 2 ml lysate was transferred to a FACS tube suitable for sonication. A Soniprep 150 set at 14 microns amplitude was used for 12 cycles of sonication (each cycle was 30 s on, 60 s off). Sonication was performed one sample at a time on ice. Following sonication, Triton-X-100 was added to the lysates (final concentration 1%), and lysates were centrifuged at $16,000 \times g$ for 12 min at 4 °C to pellet the debris. In total 50 μl of the sonicated lysate from each treatment condition was stored at −20 °C as input control. Cell lysates were pre-cleared with 100 μl of magnetic beads. Following this, the 100 μl of antibody-bead mix was then added to the cell lysates for overnight immunoprecipitation on a rotating platform at 4 °C. Following completion of this incubation, the beads were washed 5 times in 1 ml of RIPA buffer (50 mM Hepes-KOH, pH 7.5; 500 mM LiCl; 1 mM EDTA; 1% NP-40; 0.7% Na deoxycholate plus protease inhibitors as described above). The pellets were then washed in 1 ml of TBS (20 mM Tris-HCl, pH 7.6; 150 mM NaCl), centrifuged at 960 x g for 3 min at 4 °C and then, following removal of the supernatant, the ChIP samples were eluted off the magnetic beads with 200 μl of elution buffer (50 mM Tris-HCl, pH 8; 10 mM EDTA; 1% SDS) and reverse cross-linked by heating at 65 °C for 6–18 h. The beads were vortexed every 5 min for the first 15 min. A total of 200 μl of TE was added to each IP and Input samples. 8 μl of 1 mg/ml RNaseA (Ambion, 2271) was added and the samples incubated at 37 °C for 30 min. A total of 4 μl of proteinase K (20 mg/ml) (Invitrogen, 25530-049) was then added and the samples incubated at 55 °C for 1–2 h. The samples were then purified using the Qiagen MinElute PCR purification kit (28006), eluted in 30 μl elution buffer, and stored at −20 °C until required for qPCR or sequencing.

ChIP-seq libraries were made as recommended by Illumina, using the NEBNext Ultra DNA Lib Prep kit (E7370S) and NEBNext Multiplex Oligos, (Index Primers Set 2; E7500S). The quality of the libraries was assessed using the Agilent 2200 tape station (Agilent Technologies), with the Agilent D1000 ScreenTape system (Agilent, 5067-5582) and D1000 reagents (Agilent, 5067-5583). Sequencing was performed on an Illumina NextSeq 500 sequencer using the High Output kit v2 to perform 75 cycles of single-read sequencing.

Sequenced single-end ChIP-seq reads were trimmed using trim-galore (version 3.0) and aligned to the hg19 build of the human genome using Bowtie2 alignment software. Reads with a Phred quality score of <15 were removed. Non-unique or duplicate reads were removed using samtools and Picard tools (v1.98) respectively. Regions of BRD4 occupancy were determined using the SICER algorithm with a redundancy threshold of 1, a window size of 200, a fragment size of 150, an effective genome fraction of 0.75, a gap size of 200 and an FDR corrected p-value of 0.01 (suited to broader BRD4 peaks). To reduce experimental bias, a robust peak set, defined as being present in both replicates using the bedtools intersect function, was used. Regions of p53 occupancy were determined using the Macs tool (version 1.4) with a p-value cutoff of 1e-5 (suited to narrower peaks), then robust peaks that were present in at least 2 replicates were identified using the bedtools intersect function. BRD4 and p53 peaks were visualized using either the UCSC web interface or the WashU browser.

**Ectopic expression and gene knockdown by shRNA or CRISPR.** Gene knockdown or overexpression in OCI-AML3 cells was achieved by viral transfection. DNA constructs were used to generate virus for infection of OCI-AML3 cells to generate knock-down or over expression (pTRIPZ and pLKO.1 for knock-down, pLenti-CRISPR V2 for CRISPR knock-out, and pLenti4 and pCDH for over expression (Supplementary Data 5)). To generate virus, $2 \times 10^6$ HEK-293T (ATCC, CRL-11268) or HEK 293FT (gifted from Dr David Bryant, Beatson Institute for CRUK) were plated one day prior to transfection at 2 million cells per plasmid in a 10 cm plate in culturing media (DMEM with 10% v/v FBS and L-glutamine 2 mM and penicillin streptomycin 100 units per ml). Transfection was carried out in antibiotic-free culturing media (6 ml) using 10 µl lipofectamine (Lipofectamine ® 2000Reagent, 1 mg/ml, Invitrogen P/N 2887) in 400 µl Opti-mem media (Gibco Opti-mem reduced serum media, Life Technologies 31985062). This was combined with a further 400 µl Opti-mem containing 2.5 µg of plasmid DNA, 1.86 µg of ps PAX2 and 1 µg of VSVG, incubated for 20–25 min at room temperature, and then added dropwise onto the target plate of HEK-293T. After 6 h the medium was aspirated and re-freshed with 6mls of normal culture medium containing antibiotics as above. Two days after transfection of the HEK-293T, infection of the target cells was initiated by spinning down $1.5 \times 10^6$ target cells per sample and re-suspending in 400 µl of culture media in a 12-well plate. 6 ml of virus-containing media was removed from the plate of 293 T cells (the 293 T cells were refreshed with 6mls of antibiotic-containing media to allow a second infection the following day), filtered through a 0.45 µM filter and mixed with polybrene (8 µg/ml (Millipore TR-1003). To each well of target cells, 1 ml of virus-containing media was added. A GFP-expressing virus and a 'cells only' well were included as controls for the transfection and infection respectively. The 12-well plate was then spun at 2500 rpm at room temperature for 90 mins and incubated at 37 °C for 3–6 h. Cells were then transferred to a 25 cm flask with 9mls of media for an overnight incubation. The following day a second infection was carried out as described above, and 24 h after this the cells were selected with 1 µg/ml puromycin.

**Immunoblotting and antibodies.** Cells were lysed directly in 1x Laemmli sample buffer (2% SDS, 10% glycerol, 0.01% bromophenol blue, 62.5 mM Tris, pH 6.8) and boiled for 4 min. A total of 30 µg of protein was loaded into each lane of the gel. The polyacrylamide gels (BioLegend 456-1095 or NOVEX) were run at 150 V in 1x running buffer (25 mM Tris, 192 mM glycine, 0.1% SDS, pH8.3). Proteins were transferred onto PVDF membranes in transfer buffer (20% methanol, 25 mM Tris, 192 mM glycine, 0.01% SDS, 20%) for 1 h 25 min at 60 V at room temperature. Membranes were dried on top of filter paper and reactivated with methanol, blocked in TBS with 5% milk or 4% BSA and probed with antibodies overnight at 4 °C. β-Actin was used as a loading control for all western blots (1 in 200,000 dilution, Sigma A1978). All antibodies were diluted in 5% BSA TBS pH 7.5 with 0.05% sodium azide. Primary antibodies to the following targets were used at 0.1–1 µg/ml: Actin (Sigma, A1978, RRID:AB_476692), CDKN1A (Santa Cruz, 397, RRID:AB_632126), c-MYC (Santa Cruz, 764, RRID:AB_631276), MDM2 (Santa Cruz, 965, RRID:AB_627920), PARP (Cell Signalling, 9542, RRID:AB_2160739), BBC3 (Imgenex, 458, RRID: AB_1151450), TP53 (DO1) (Santa Cruz, 126, RRID:AB_628082), NOXA (Abcam, 13654, RRID:AB_300536), BRD4 (Abcam, 128874, RRID:AB_11145462), BCL-2 (Cell signaling, 2870, RRID:AB_2290370) (Supplementary Data 6). The following morning membranes were washed in TBS pH 7.5 (3 ×5 min washes) and then incubated with species-matched HRP-linked secondary antibody. The following secondary antibodies were added for a 1-h incubation at room temperature: anti-mouse IgG HRP-linked (Dako, p0447) and anti-rabbit IgG HRP-linked (Cell signaling, 7074 s), both at a dilution of 1:5,000. Following incubation with secondary antibodies, membranes were given 3 ×10-min washes in TBS pH7.5 + 0.1%Tween 20 and then incubated in ECL Western blotting chemiluminescent substrate (Thermo Scientific, 32106) for 1 min. Visualisation of protein bands was then carried out using either GE Healthcare Amersham hyperfilm on a Kodak X-Omat 480 RA X-ray processor, or BIORAD ChemiDoc Imaging system. All experiments were performed multiple times and confirmed to be reproducible.

**FACS analysis.** For flow cytometric evaluation of apoptosis in AML cell lines and primary murine AML cells treated with the above single agents or combination treatments, apoptosis was measured by combined Annexin-V/propidium iodide (PI) staining (Cambridge Bioscience, K101-400), myeloid differentiation was assessed using stains for CD11b (eBioscience, 12-0081) and GR1 (eBioscience, 25-5931), lymphoid differentiation was assessed using stains for CD4 (Biolegend, 100553), CD8a (eBioscience, 12-0081), CD19 (eBioscience, 47-0193080) and B220 (eBioScience, 45-0452). Drug or vehicle-treated cells were collected and first washed with 1 ml of PBS. Cells were then re-suspended in 1 ml of 1× Binding Buffer. Following this, the relevant stain of interest was added and samples incubated for 10 min at room temperature in the dark. Samples of human cell lines were then analyzed on a FACSCalibur (BD Biosciences) and primary murine samples were analyzed on a FACSCanto (BD Bioscience). Cells were centrifuged at 1000 rpm for 5 min and resuspended in 1 ml sorting buffer (PBS 2 % FBS) and DAPI was added immediately prior to sort (final concentration 1 µg/ml; 1 in 1000 dilution). Cells were sorted by Jennifer Cassells at the Paul O' Gorman Leukaemia Research Centre using a BD FACSARIA III sorter and were collected in 1 ml sorting buffer, pooled, centrifuged as above and left to recover in culturing medium at 37 °C. Gating strategy for sorting GFP Positive cells was live cells>cell profile>single cells>GFP positive cells.

For assessment of disease engraftment and disease burden in mice, blood was collected via tail vein. Red blood cell lysis was carried out using BioLegend 10x red blood cell lysis buffer as per manufacturers' instructions (incubation time was increased to 8 min after optimization). After centrifugation, cell pellets were resuspended in 400 µl PBS and % GFP+ was measured using the BD Fortessa flow cytometer. At mouse cull, leg bones were cleaned upon removal then crushed using a pestle and mortar. The cell suspension was filtered and washed with ice-cold PBS + 2% FBS. For spleen and thymus, the organ was placed in a 70 µM cell strainer and pushed through using the plunger of a 5 ml syringe into a 50 ml falcon containing PBS + 2% FBS. Suspensions were pelleted at 200 g for 5 min, and the pellets resuspended in red cell lysis buffer before being pelleted as above. Finally, cells were washed in 1 ml PBS + 2% FBS. For FACS, compensation was calculated using ULTRACOMP beads (E biosciences 01-2222-41) incubated for 20 min with each antibody used in the experiment. Fluorescence minus one (FMO) controls were also included using cells incubated with a mix of all antibodies except one in each FMO control. A master mix was prepared containing all antibodies at a 1 in 250 dilution (myeloid cells [CD11b, E- bioscience, 17-0112-83; Gr1, E- bioscience, 25-5931-81], B cells [B220, E- bioscience, 45-0452-80; CD19, E- bioscience, 47-0193080], T cells [CD4, Biolegend, 100547; CD8, E- bioscience, 12-0081-82] (Supplementary Data 6)) in PBS + 2% FBS and cell pellets were re-suspended in 50 µl of master mix for a 30 min incubation on ice and away from light. Immediately prior to FACS analysis DAPI was added at a 1 in 1000 dilution (1 µg/ml) (Sigma, D9542-1MG).

**Statistical analysis.** Results were statistically analyzed in GraphPad Prism (version 7.0, GraphPad Software Inc., San Diego, CA) and presented as mean±SEM. Statistically significant differences between two groups were assessed by two-tailed unpaired t-test. $p \leq 0.05$ was considered significant. ns, not significant; *$p \leq 0.05$; **$p \leq 0.01$; ***$p \leq 0.001$; ****$p \leq 0.0001$ for indicated comparisons. Statistical details of each experiment can be found in the Results and Figure Legend sections.

**Reporting summary.** Further information on experimental design is available in the Nature Research Reporting Summary linked to this paper.

## Data availability
Source Data Files are available in the Supplementary Data as an excel file with each sheet being a separate figure: "Figure Data.xls". For Figures with associated raw data, see the Supplementary Dataset File. Chip Seq and RNA seq data has been deposited in the Gene Expression Omnibus (GEO) under accession codes GSE132246, GSE132247, GSE132244, and GSE 132245. All other data supporting the findings of this study are available from the corresponding author upon reasonable request. Links to GEO accession are below (Supplementary Data 7): https://www.ncbi.nlm.nih.gov/geo/query/acc.cgi?acc=GSE132244, https://www.ncbi.nlm.nih.gov/geo/query/acc.cgi?acc=GSE132245, https://www.ncbi.nlm.nih.gov/geo/query/acc.cgi?acc=GSE132246, https://www.ncbi.nlm.nih.gov/geo/query/acc.cgi?acc=GSE132247. Source data are provided with this paper.

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

## Acknowledgments

We thank Robert J. Sims, III and Jennifer A Mertz of Constellation Pharmaceuticals Inc. for providing CPI0610 and advice on in vivo studies and the staff of BICR Biological Services Unit for assistance with mouse experiments. Work in the lab of P.D.A. was funded by CRUK program grant C10652/A16566. We thank support from Children with Cancer UK and the Howat Foundation (to Ka.Ke. and J.C.). Kr.Ki. was funded by Wellcome Trust (Grant number 105641/Z/14/Z). H.J.S.S. was supported by the Sussex Cancer Fund. Additional funding from CRUK Glasgow Centre (A25142) and Core Services at the Cancer Research UK Beatson Institute (A17196). This study was supported by the Glasgow Experimental Cancer Medicine Centre, which is funded by Cancer Research UK and the Chief Scientist's Office, Scotland. Experiments in the lab of T.J.T.C. were supported by Sussex Cancer Fund. X.H. is a John Goldman Fellow [Leuka 2016/JGF/0005].

## Author contributions

Contributions to the manuscript: A.-L.L., P.D.A., and M.C. conceived the idea for the project; A.-L.L. and A.N. conducted most of the experiments; S.L. performed additional in vitro experiments; M.T., L.R.D., J.C., and C.R. assisted with in vivo experiments; H.J.S.S. and T.J.T.C. performed and supervised, respectively, experiments with primary human AML blasts; L.M. performed cell viability assays to calculate combination indices; Kr.Ki., J.L., S.W.G.T., and M.C. provided technical assistance or advice; J.-I.S and K.M.R. gifted reagents and shared critical unpublished data; W.C. performed RNA- and ChIP-sequencing; C.M. performed sequencing of TP53 in human AML blasts; K.G., N.A.R., X.L., and J.C. conducted bioinformatic analyses; X.H. assisted and advised on the MLL-AF9 mouse model; S.A.A. and T.H. assisted in procuring drugs for in vivo experiments; B.X.H. gifted RG7112; Ka.Ke., J.P.M. and K.B. advised and supervised in vivo work; A.-L.L., A.N., S.L., and P.D.A. wrote the MS. P.D.A. supervised the project. All authors have reviewed and approved the manuscript.

## Competing interests

M.C. is Chief Investigator of an investigator-led clinical trial (EudraCT number 2018-001843-29) funded by CR-UK (grant number A24896: CRUK/17/016) using idasanutlin (RG7388) in chronic myeloid leukemia. The remaining authors declare no competing interests.
