## [Peer Review File · Nature Communications]

Reviewers' comments:

Reviewer #1 (Remarks to the Author); expert in leukaemia and p53:

A large fraction of AMLs have wild-type p53 and can therefore respond to MDM2 inhibitor treatment. This was known already and much of the mechanisms by which such drugs kill leukemic cells is also known. Moreover, it has been shown that BET inhibitors can kill leukemic cells and much is known about that mechanism of killing >> induction of the pro-apoptotic BH3-only protein BIM (Z Xu, Leukemia, 2016).

The authors claim that BET inhibitors and MDM2 inhibitors SYNERGISE in the killing of AML cells. There are very clear assays to demonstrate synergy rather than simple additivity of drug combinations in killing malignant cells - i.e. checkerboard titration of both drugs against each other in 2-fold dilutions up and down from the IC50, followed by BLISS analysis. This was not done in this manuscript! To me the data indicate that BET inhibitors and MDM2 inhibitors are simply additive but not synergistic in the killing of AML cells. Moreover, the mechanisms of the cooperation of these two treatments in the killing of AML cells is not clearly elaborated (see above). Finally, here are already several prior publications demonstrating the mechanisms by which MDM2 inhibitors kill cancer cells, the mechanisms by which BET inhibitors kill cancer cells (Z Xu, Leukemia 2016). Due to these papers, the data shown here are not of sufficient novelty to warrant publication in a leading journal.

detailed comments:

1) p53 gene sequence analysis was conducted on only a minority of the AML patient samples; this should have been performed on all samples

2) The MDM2 inhibitor RG7112 binds much more potently to human MDM2 (called HDM2) than to mouse MDM2; thus the in vivo studies conducted greatly under-estimate on-target toxicity of this compound. This is a problem with the data presented.

The authors use different MDM2 inhibitors in their work without explaining why; most likely to avoid in vivo toxicity to healthy cells.

3) all Western blots lack molecular weight markers

4) in many of the AML cell lines analyses there clearly is not strong synergy between the MDM2 inhibitor and the BET inhibitor (see above) >> also the mechanisms of drug cooperatively in the killing of AML cells should have been examined in at least three AML cell lines (showing different levels of drug cooperatively) and not just one cell line

5) in the in vivo experiments with the MLL-AF9 AML there was only minor (unimpressive) extension of lifespan with the MDM2 inhibitor + BET inhibitor combination compared to treatment with single agents

6) Figure 3: the functionality of BCL-2 over-expression was not proven. This, according to many papers, should render the cell lines resistant to the MDM2 inhibitor

7) Figure 3: discovery that loss of Puma and NOxa renders cells resistant to MDM2 inhibitor induced killing is not novel >> it is therefore a given that their loss will also protect cells from the combination of MDM2 inhibitor + BET inhibitor treatment, at least partially.

Reviewer #2 (Remarks to the Author); expert in leukaemia and BET inhibitors:

In this manuscript Latif and colleagues demonstrate that a combination of inhibition of both MDM2 and the BET protein BRD4 has synergistic activity against a range of AML genotypes, both in vitro and in vivo. They further demonstrate a degree of transcriptional synergy at p53 target genes for the combination and exclude a number of mechanisms for the synergy. Finally, they present preliminary evidence that the synergy may relate to an inhibition of BRD4-mediated repression of p53 at these target genes.

This is an interesting manuscript and the translational work detailing the combined therapy is compelling and highly promising. However, although the researchers quite convincingly present data to refute various mechanisms for the synergy between MDM2 and BET inhibition, the data that they provide to support their preferred mechanism, that BRD4 represses transcription of TP53 targets and that dual inhibition synergises to both stabilise TP53 protein and through relief of repression further activates it, although plausible, is not compellingly justified. Further work is therefore required to substantiate these mechanistic claims and round off the paper.

I have the following specific points that the authors should address;

Major

1. As detailed above, the data suggesting that BRD4 represses transcription of p53 targets is currently weak and should be further substantiated.
2. For example, does BRD4, particularly the short-form linked previously with transcriptional repression bind directly to TP53 in the cellular context of OCI-AML3, the major cell line work-horse for this project?
3. The authors should also perform further experiments to document transcriptional repression – only repression of two genes CDKN1A and PUMA are shown, when BRD4S is overexpressed. What about other candidate genes? What about the two candidates suggested to be important for efficient synergistic killing of AML cell lines, BBC3 and NOXA?
4. Similarly, can the authors show in luciferase reporter assays that TP53 mediated transactivation is abrogated by overexpression of BRD4S?
5. How is this this repression mediated? Is this through recruitment of co-repressors and/or the loss of co-activators?

Other points

1. For Figure 1B, can the TP53 status be determined for the other samples. This would be helpful to correlate with the functional synergy.
2. Figure 2G. Please print the proportion of GFP cells larger
3. What is meant by synergistic up-regulation? Does this have a statistical basis or is it entirely arbitrary? How do these compare with the Venn Diagrams in Fig S3 C + D?

We thank the reviewers for their constructive critiques. In response to their concerns, we have performed additional experiments and analyses and revised the text to improve the MS. Our major conclusions are unchanged – that the combination of MDM2i and BETi is synergistically lethal to AML due to concerted activation of p53 by MDM2i's ability to stabilize p53 and BETi's ability to abrogate BRD4-mediated repression of p53 target genes. This is a novel and unanticipated mechanism of drug synergy with broad potential implications for AML therapy, since most AML retain wild type p53.

We have added new Figure 1P, Supplementary Figures S1E-J, S1N-S, S5, S6H and S7. We also include for the reviewers only, Reviewers' Figures 1-3. These new data show –

1. Drug combination (MDM2i+BETi) synergy scores by the Bliss method, for 3 wild type and 3 mutant TP53 AML cell lines (Figure S1E-J). In addition, drug combination (MDM2i+BET PROTAC degraders) synergy scores by the Bliss method, for 3 wild type TP53 AML cells lines (Figures S1N-S).
2. Gene expression data showing a role for BET family proteins as repressors of p53 target gene expression (Figure S5)
2. Additional analysis of ChIP-seq data showing binding of BRD4 to p53 target genes (Figure S6H).
3. Repression of p53 target genes by ectopically expressed BRD4 in AML3 and MOLM13 cells (Figure S7).
4. Toxicity dose-response of MDM2i and BETi in 3 wild type and 3 mutant TP53 AML cell lines (Reviewers' Figure 1).
5. Data supporting a role for histone methyltransferase G9a in BRD4-mediated repression of p53 target genes (Reviewers' Figure 2).
4. Data confirming that the level of expression of BCL2 is not a major determinant of sensitivity to MDM2i (Reviewers' Figure 3).

Notable text changes or text referred to in this response are colored blue in the MS.

Reviewer #1 (Remarks to the Author); expert in leukaemia and p53:

A large fraction of AMLs have wild-type p53 and can therefore respond to MDM2 inhibitor treatment. This was known already and much of the mechanisms by which such drugs kill leukemic cells is also known. Moreover, it has been shown that BET inhibitors can kill leukemic cells and much is known about that mechanism of killing >> induction of the pro-apoptotic BH3-only protein BIM (Z Xu, Leukemia, 2016).

The reviewer is correct that a substantial literature addresses the mechanism of killing by BETi and MDM2i as single agents. We omitted to cite some of this work and have now cited Xu et al., 2016 [1] in the Results section (page 9). However, our study concerns the mechanism of AML cell killing by the MDM2i/BETi combination. It should not be assumed that the mechanism of killing by a synergistic drug combination is the same as killing by one of the drugs on its own. The major conclusion of our study is that the MDM2i/BETi combination synergize by upregulating p53 (MDM2i) and relieving BRD4-mediated repression of p53 target genes (BETi). This is a novel mechanism of synergy that was not predicted based on the single drug mechanisms.

The authors claim that BET inhibitors and MDM2 inhibitors SYNERGISE in the killing of AML cells. There are very clear assays to demonstrate synergy rather than simple additivity of drug combinations in killing malignant cells - i.e. checkerboard titration of both drugs against each other in 2-fold dilutions up and down from the IC50, followed by BLISS analysis. This was not done in this manuscript! To me the data indicate that BET inhibitors and MDM2 inhibitors are simply additive but not synergistic in the killing of AML cells.

We showed in Figure 1E-J the calculated CI (Combination Index) for the drugs, using the Chou and Talalay method [2]. A CI<1,=1,>1 indicates drug synergy, additivity or antagonism, respectively. The CIs of wild type TP53 cell lines OCI-AML3, MOLM-13, and MV4-11 are all substantially <0, 0.34, 0.21 and 0.29 respectively (Figure 1E-G), over a range of drug dose ratios (Figure S1B-D). In contrast, all three mutant TP53 cell lines have CIs>1 (Figure 1H-J). We explained our Results at bottom of page 5 and top of page 6.

However, to more quantitatively address the reviewer's concern, we used the Bliss method [3]. To do this, first we determined the IC50 of each drug in 6 AML cell lines: 3 wild-type TP53 cell lines (OCI-AML3, MOLM13 and MV411) and 3 mutant TP53 cell lines (KASUMI-1, KG1a and THP1). As expected, MDM2 inhibitor Nutlin-3 depends on wild-type TP53 to induce dose-dependent cell death. In contrast, BET inhibitor CPI203 causes cell death in a p53-independent manner (Reviewers' Figure 1A, B). We then performed checkerboard

titration of both drugs against each other in 2-fold dilutions and used the Bliss method to calculate synergy [3](Supplementary Figure 1E-J). All 3 wild-type *TP53* cell lines showed a higher average Bliss synergy score (6.3, 3.6 and 10.3 for OCI-AML3, MOLM13 and MV411, respectively) compared to all 3 mutant *TP53* cell lines (2.6, 2.7 and 1.5 for KASUMI-1, KG1a and THP1, respectively). Most synergistic area Bliss scores were also typically much higher in wild-type *TP53* AML (12.0, 11.2 and 25.3 for OCI-AML3, MOLM13 and MV411, respectively), compared to mutant *TP53* AML (5.7, 6.4, and 3.8 for KASUMI-1, KG1a and THP1, respectively). Note also that each *TP53* mutant cell line shows a Bliss synergy score <0 indicative of antagonism at several dose combinations; this is barely observed for the three *TP53* WT cell lines. In sum, both the Chou and Talalay method [2] and the Bliss method [3] confirm that the drug combination is synergistically lethal in wild type *TP53* AML, and much less so – or not at all - in mutant *TP53* AML. Furthermore, we used small molecules BET PROTAC degraders BETd-260 or ARV-825 in combination with MDM2i. Beyond consistency with BETi, our results show exceptionally synergistic effects in wild-type *TP53* cell lines (Figure 1P, Supplementary Figure S1N-S). For example, in AML3 cells, most synergistic area Bliss scores are 34.0 for MDM2i with BETd-260 (Figure 1P) and 32.1 for MDM2i with ARV-825 (Supplementary Figure S1Q). This indicates that BET PROTAC together with MDM2i can also represent a promising therapeutic agent for AML.

Moreover, the mechanisms of the cooperation of these two treatments in the killing of AML cells is not clearly elaborated (see above). Finally, here are already several prior publications demonstrating the mechanisms by which MDM2 inhibitors kill cancer cells, the mechanisms by which BET inhibitors kill cancer cells (Z Xu, Leukemia 2016). Due to these papers, the data shown here are not of sufficient novelty to warrant publication in a leading journal.

As noted above, we proposed a novel mechanism of synergy between MDM2i and BETi, not considered in previous reports. To recap - we showed that in AML cells BRD4 binds directly to p53 target genes and represses expression of those genes. Therefore, full activation of p53 and expression of its pro-apoptotic target genes, such as NOXA and BBC3, is achieved through synergy of MDM2i and BETi, whereby the former stabilizes p53 and the latter displaces repressive BRD4 from p53 target genes.

More recent results have extended this model. BRD4 and G9a have been reported to physically interact [4-6]. Moreover, it has been previously reported that BRD4 can suppress expression of autophagy genes through G9a [4]. Therefore, we hypothesized that knockdown of G9a or inhibition of G9a would also synergize with MDM2i. We first tested our hypothesis by knock down of G9a with two different shRNAs in the OCI-AML3 cell line (Reviewers' Figure 2A). We observed dramatic killing of AML3 cells lacking G9a in the presence of MDM2i, consistent with the idea that G9a promotes repression of p53 apoptotic genes (Reviewers' Figure 2B, C), similar to BRD4. We also treated cells with G9a inhibitor UNC0631 together with MDM2i to test whether there is synergy between them in 3 wild-type *TP53* cell lines (OCI-AML3, MOLM13 and MV411). For each cell line, the IC50 of UNC0631 was in the mM range (Reviewers' Figure 2D). Consistent with shRNA perturbation data and our model, we found the G9a inhibitor synergizes with MDM2i to kill AML cells (Reviewers' Figure 2E-G), CIs of 0.48, 0.69 and 0.64 for OCI-AML3, MOLM13 and MV411, respectively. These data and the reported physical interaction between G9a and BRD4 [4-6] are consistent with a model whereby G9a is a co-repressor role in BRD4-mediated repression of p53 target genes. These prior reports have been noted in the Discussion (page 14). However, since mechanistic studies into BRD4/G9a as a repressor of p53 target genes are ongoing, we prefer to exclude these data from the current MS.

Detailed comments:

1) *p53 gene sequence analysis was conducted on only a minority of the AML patient samples; this should have been performed on all samples*

We agree that ideally we would have sequence data on all the samples. However, due to limited DNA sample availability from the clinical specimens, this was not possible. However, as expected from our model, the majority of the samples showed a trend towards greater than additive killing by the combination (x axis < 1), compared to the single drugs. Of the 4 samples for which *TP53* mutation status was determined by DNA sequencing, all retained wild-type *TP53* and of these 3 showed at least a trend towards synergy (Figure 1B). The text in the Results section has been amended to note this limitation (page 5).

2) *The MDM2 inhibitor RG7112 binds much more potently to human MDM2 (called HDM2) than to mouse MDM2; thus the in vivo studies conducted greatly under-estimate on-target toxicity of this compound. This is a problem with the data presented.*

On-target toxicity is due to MDM2i inhibiting MDM2 in normal non-neoplastic tissues. This, in turn, depends on the “therapeutic window”, i.e. finding a dose of the drug that kills AML without on-target toxicity in normal tissues. *In vivo* on-target toxicity against human MDM2 (HDM2) is something that can only be addressed in humans in a well-designed clinical study, to see if the efficacy translates in the context of tolerability of the combination in patients. Prior to human studies, toxicology data is generated in GLP studies utilizing appropriate species for extrapolation to humans (for RG7112 this was the Wistar rat and Cynomolgus Monkey [7]). Modelling on-target toxicity in humans is a known limitation of most oncology animal model systems, and not one that we could reasonably address here.

We considered the reviewer’s concern together with Roche (co-author Brian Higgins) before initiating these experiments. We elected to use an MDM2i, RG7112, that has been tested clinically in humans [8, 9]. RG7112 was used here for the studies in murine models, rather than the human clinical lead molecule, Idasanutlin (RG7388), because Idasanutlin is even more optimized for human specificity than RG7112 [10]. Since the goal of the experiments was to look at anti-tumor activity in murine-based models (i.e. against murine AML cells), RG7112 was felt to be the better representative Nutlin MDM2 antagonist for this model system, whereas Idasanutlin would be better for a human (xenograft/PDX) model system. Doses of RG7112 used were in ranges previously demonstrated as tolerable and free of toxicities in mouse efficacy models [11]. The Discussion text has been modified to note this caveat regarding on-target toxicity (page 15).

The authors use different MDM2 inhibitors in their work without explaining why; most likely to avoid in vivo toxicity to healthy cells.

We used Nultin-3 in initial expts in human AML cell lines, as one of the original benchmark MDM2i [12]. For *in vivo* studies we preferred to use a clinical grade molecule and opted for RG7112, partly for reasons outlined above and partly because two of our authors (TH and MC) already had an MTA with Roche (author BH) to facilitate the transfer of the drug. Roche donated the drug cost-free under MTA as a collaboration; otherwise the cost would have been prohibitive for these *in vivo* experiments. We did not pick different MDM2i to avoid toxicity. The rationale for RG7112 has been noted in the Results section (page 7).

3) *all Western blots lack molecular weight markers*

We added as requested.

4) *in many of the AML cell lines analyses there clearly is not strong synergy between the MDM2 inhibitor and the BET inhibitor (see above) >> also the mechanisms of drug cooperatively in the killing of AML cells should have been examined in at least three AML cell lines (showing different levels of drug cooperatively) and not just one cell line.*

We have now used both the Chou-Talalay method (CI, Combination Index) and the Bliss method to evaluate synergy [2, 3]. CIs were calculated in all 6 AML cell lines (Figure 1E-J). The CIs of the wild type *TP53* cell lines OCI-AML3, MOLM13 and MV411 are 0.34, 0.29, and 0.21 respectively, all robustly synergistic. We also used Bliss method to calculate synergy as requested (Figure S1E-J). The mean Bliss scores of the same wildtype *TP53* cell lines are 6.35, 3.59 and 10.31 respectively, and peak Bliss scores are >10-20. Although drug synergy appears to be typical of *TP53* wild type AML, given the molecular heterogeneity of AML (Klco et al., 2013; Papaemmanuil et al., 2016), dominant synergy mechanisms are likely to vary, as suggested by the reviewer. Although we have only been able to assess the mechanism of synergy in one cell line, OCI-AML3, the genotype of these cells (*NPM1* and *DNMT3A* mutant and *TP53* wild type) is characteristic of ~10-15% of AML. The text has been amended to note this caveat (page 15).

5) *in the in vivo experiments with the MLL-AF9 AML there was only minor (unimpressive) extension of lifespan with the MDM2 inhibitor + BET inhibitor combination compared to treatment with single agents*

The MLL-AF9 leukemia model is much more aggressive than the Trib2 model and, at this dose of leukemic cells, the vehicle-treated mice succumbed to leukemia in less than 2 weeks (to some extent the rate of disease progression is simply a function of the number of injected cells). Most important, in this model the 50% survival with combination is substantially greater than additive (6, 8, 8 and 15 days, for vehicle, CPI0610 (+2 days), RG7112 (+2 days) and combination (+9 days), respectively) (Figure 2K). The extension of lifespan is very significant with p value <0.0001 (for combo treatment compared to single drugs alone). The text has been modified to emphasize this point (page 8).

6) *Figure 3: the functionality of BCL-2 over-expression was not proven. This, according to many papers, should render the cell lines resistant to the MDM2 inhibitor.*

As noted by the reviewer, in Figure 3C, D we did not observe significant resistance to MDM2i conferred by BCL2 ectopic expression. To verify this, we assessed the effect of BCL2 protein over-expression on sensitivity to MDM2i (Nutlin-3) in three wild type *TP53* cell lines (OCI-AML3, MOLM13 and MV411) at a range of Nutlin-3 doses. Consistent with our previous observation (Figure 3C-D), OCI-AML3 cells overexpressing BCL2 did not show any significant difference in sensitivity to MDM2i compared to control cells (Reviewers' Figure 3B). Similarly, Molm13 and MV4-11 overexpressing BCL2 did not show any significant difference in sensitivity to MDM2i either (Reviewers' Figure 3C-D). Other observations also challenge the idea that ectopic expression of BCL2 should be sufficient to confer resistance to MDM2i. First, Michael Andreef's group assessed the correlation between BCL2 protein expression and sensitivity to Nutlin-3a in 14 representative AML cell lines, including OCI-AML3, MV4-11 and MOLM-13 used in our study [14]. They found that "*there were also no correlations between the sensitivities to Nutlin-3a and and expression of any pro-survival BCL-2 protein*". Second, analysis of data in the BeatAML database confirms that there is no direct correlation between expression of BCL2 and resistance to Nutlin-3a. On the contrary, there is a modest but significant inverse correlation (Reviewer's Figure 3E). Third, even in the landmark study showing synergy between MDM2i and BCL2i in AML, also from Andreef's lab [13], the authors do not show that ectopic expression of BCL2 renders cell lines resistant to MDM2i (as far as we can tell). In sum, we are not aware of good evidence that elevated BCL2 confers resistance to MDM2i, as suggested by the reviewer. However, the reviewer is correct to point out that "negative data" as shown in Figure 3C, D cannot rigorously exclude a role for downregulation of BCL2, and the text has been modified to note this (page 9). However, these negative data can prompt a search for alternative mechanisms to be validated by "positive data", as was the case here.

7) *Figure 3: discovery that loss of Puma and NOxa renders cells resistant to MDM2 inhibitor induced killing is not novel >> it is therefore a given that their loss will also protect cells from the combination of MDM2 inhibitor + BET inhibitor treatment, at least partially.*

We agree that this result is not particularly surprising. It is not the main point of the manuscript. However, it should not be assumed that the mechanism of killing by a synergistic drug combination is the same as killing by one of the drugs on its own. Indeed, if two drugs as single agents have distinct killing mechanisms (A and B), as is generally thought to be the case for BETi and MDM2i, it follows that the combination might kill by enhanced A only, B only, both A and B or another mechanism. Thus, it was important to test the consequence of PUMA+NOXA inactivation (Figure 3O-P).

Reviewer #2 (Remarks to the Author); expert in leukaemia and BET inhibitors:

In this manuscript Latif and colleagues demonstrate that a combination of inhibition of both MDM2 and the BET protein BRD4 has synergistic activity against a range of AML genotypes, both in vitro and in vivo. They further demonstrate a degree of transcriptional synergy at p53 target genes for the combination and exclude a number of mechanisms for the synergy. Finally, they present preliminary evidence that the synergy may relate to an inhibition of BRD4-mediated repression of p53 at these target genes.

This is an interesting manuscript and the translational work detailing the combined therapy is compelling and highly promising. However, although the researchers quite convincingly present data to refute various mechanisms for the synergy between MDM2 and BET inhibition, the data that they provide to support their preferred mechanism, that BRD4 represses transcription of TP53 targets and that dual inhibition synergises to both stabilise TP53 protein and through relief of repression further activates it, although plausible, is not compellingly justified. Further work is therefore required to substantiate these mechanistic claims and round off the paper.

I have the following specific points that the authors should address;

Major

1. *As detailed above, the data suggesting that BRD4 represses transcription of p53 targets is currently weak and should be further substantiated.*

Previously, gene expression data in the MS focused on upregulation of p53 target genes by the MDM2i/BETi combination (Figure 3F-L). To substantiate the role of BRD4 as a repressor, we have more closely analyzed gene expression changes after BETi alone. Remarkably, p53 was ranked as one of the top upstream

regulators of genes differentially expressed by CPI203 alone (Supplementary Figure 5A). Focused analysis of p53 target genes directly showed that CPI203 alone was sufficient to increase expression of many p53 target genes in OCI-AML3 cells (Supplementary Figure 5B, C). For several p53 target genes this was confirmed by qPCR after treatment with CPI203 or knock down of BRD4 by lentivirus-encoded shRNA (Supplementary Figure 5D, E). Inhibition of BET family proteins with a potent PROTAC BET degrader BETd-260 strongly upregulated expression of p53 target gene p21 at the protein level (Supplementary Figure 5F, G). Together with previous data showing that BRD4 is bound to many p53 target genes (Figure 5D-F) and is able to repress some p53 target genes (Figure 5G-I and Supplementary Figure 7), this strengthens data in support of BRD4's role as a repressor of p53 target genes.

2. For example, does BRD4, particularly the short-form linked previously with transcriptional repression bind directly to TP53 in the cellular context of OCI-AML3, the major cell line work-horse for this project?

Yes, a physical interaction between BRD4 and p53 was reported by Wu et al [5] and in OCI-AML3 cells by authors of this study (HS and TC [15]). However, the region in BRD4 that interacts with p53 has been identified and it is contained within both long and short isoforms of BRD4. Therefore, we would expect that both isoforms are able to interact with p53 [5]. It should also be noted that there is a lack of data to support the idea that BRD4 short-form is preferentially repressive. While Ott and coworkers showed that BRD4S is a preferential repressor of transcriptionally silent latent HIV virus by recruitment of repressive SWI/SNF chromatin remodeling complexes [16], Chiang and coworkers reported that BRD4L participates in repression of the HPV-encoded E6 gene [17]. Also, Ryan and coworkers showed that both BRD4S and BRD4L act to repress autophagy genes by recruitment of histone methyltransferase G9A [4]. The text has been modified to remove any misleading suggestion that BRD4S is likely to be preferentially repressive.

3. The authors should also perform further experiments to document transcriptional repression – only repression of two genes CDKN1A and PUMA are shown, when BRD4S is overexpressed. What about other candidate genes? What about the two candidates suggested to be important for efficient synergistic killing of AML cell lines, BBC3 and NOXA?

BBC3 and PUMA are alternative names for the same gene (Figure 5I). In the MS, we introduced the gene as BBC3 (PUMA), but thereon used PUMA because this is much more common in the literature. We extended our analysis to NOXA, as well as another 7 p53 target genes: GDF15, GADD45a, LAPTM5, FUCA1, TP53INP1, SERTAD1 and SESN2. In all cases there was a trend to lower expression in BRD4S overexpressing cells (Supplementary Figure 7A-H). We also tested BRD4 ectopic expression in MOLM13 cells and found similar results (Supplementary Figure 7K-N).

4. Similarly, can the authors show in luciferase reporter assays that TP53 mediated transactivation is abrogated by overexpression of BRD4S?

We transduced a TP53 promoter-based dual reporter cassette (luciferase and GFP) into cells ectopically expressing BRD4S or control. Similar to the gene expression assays above, we observed lower expression in the BRD4S-expressing cells (Supplementary Figure 7I, J).

5. How is this this repression mediated? Is this through recruitment of co-repressors and/or the loss of co-activators?

As noted in response to Reviewer 1, repression is likely to be mediated by co-repressor G9A. BRD4 and G9a have been reported to physically interact [4-6]. Moreover, it has been previously reported that BRD4 can suppress expression of autophagy genes through G9a [4]. Therefore, we hypothesized that knockdown of G9a or inhibition of G9a would also synergize with MDM2i. We first tested our hypothesis by knock down of G9a with two different shRNAs in the OCI-AML3 cell line (Reviewers' Figure 2A). We observed dramatic killing of AML3 cells lacking G9a in the presence of MDM2i, consistent with the idea that G9a promotes repression of p53 apoptotic genes (Reviewers' Figure 2B, C), similar to BRD4. We also treated cells with G9a inhibitor UNC0631 together with MDM2i to test whether there is synergy between them in 3 wild-type TP53 cell lines (OCI-AML3, MOLM13 and MV411). For each cell line, the IC50 of UNC0631 was in the mM range (Reviewers' Figure 2D). Consistent with shRNA perturbation data and our model, we found the G9a inhibitor synergizes with MDM2i to kill AML cells (Reviewers' Figure 2E-G), CIs of 0.48, 0.69 and 0.64 for OCI-AML3, MOLM13 and MV411, respectively. These data and the reported physical interaction between G9a and BRD4 [4-6] are consistent with a model whereby G9a is a co-repressor role in BRD4-mediated repression of p53 target genes. These prior

reports have been noted in the Discussion (page 14). However, since mechanistic studies into BRD4/G9a as a repressor of p53 target genes are ongoing, we prefer to exclude these data from the current MS.

Other points

1. For Figure 1B, can the TP53 status be determined for the other samples. This would be helpful to correlate with the functional synergy.

We agree that ideally we would have sequence data on all the samples. However, due to limited DNA sample availability from the clinical specimens, this was not possible. However, as expected from our model, the majority of the samples showed a trend towards greater than additive killing by the combination (x axis < 1), compared to the single drugs. Of the 4 samples for which TP53 mutation status was determined by DNA sequencing, all retained wild-type TP53 and of these 3 showed at least a trend towards synergy. The text in the Results section has been amended to note this limitation (page 5).

2. Figure 2G. Please print the proportion of GFP cells larger

We added the info as requested.

3. What is meant by synergistic up-regulation? Does this have a statistical basis or is it entirely arbitrary? How do these compare with the Venn Diagrams in Fig S3 C + D?

Genes synergistically up-regulated (Up) by the drug combination were rigorously identified as those where $C/(A+B) \Rightarrow 1.25$ and combination FPKM $>$ DMSO FPKM, and synergistically down-regulated genes (Down) where $C/(A+B) \Rightarrow 1.25$ and combination FPKM $<$ DMSO FPKM (where C= combination FPKM – DMSO FPKM; A= nutlin-3 FPKM – DMSO FPKM; B= CPI203 FPKM – DMSO FPKM). This is included in the legend to Figure 3E.

This analysis yielded 252 genes that were synergistically up-regulated and 94 genes that were synergistically down-regulated by the drug combination (Figure 3E and Supplementary Tables 1 and 2). Supplementary Figures 3C and D show the numbers of significantly ($p < 0.05$) differentially expressed genes from each treatment, single treatments and combination compared to vehicle. Figure 3C indicates that 356 genes are upregulated by the combination but neither single treatment alone. However, compared to the 252 synergistically upregulated genes, the 356 genes does not include an effect size threshold ($\Rightarrow 1.25$).

References

1. Xu, Z., et al., *BET inhibition represses miR17-92 to drive BIM-initiated apoptosis of normal and transformed hematopoietic cells*. *Leukemia*, 2016. **30**(7): p. 1531-41.
2. Chou, T.C. and P. Talalay, *Quantitative analysis of dose-effect relationships: the combined effects of multiple drugs or enzyme inhibitors*. *Adv Enzyme Regul*, 1984. **22**: p. 27-55.
3. Ianevski, A., A.K. Giri, and T. Aittokallio, *SynergyFinder 2.0: visual analytics of multi-drug combination synergies*. *Nucleic Acids Res*, 2020.
4. Sakamaki, J.I., et al., *Bromodomain Protein BRD4 Is a Transcriptional Repressor of Autophagy and Lysosomal Function*. *Mol Cell*, 2017. **66**(4): p. 517-532 e9.
5. Wu, S.Y., et al., *Phospho switch triggers Brd4 chromatin binding and activator recruitment for gene-specific targeting*. *Mol Cell*, 2013. **49**(5): p. 843-57.
6. Dawson, M.A., et al., *Inhibition of BET recruitment to chromatin as an effective treatment for MLL-fusion leukaemia*. *Nature*, 2011. **478**(7370): p. 529-33.
7. Glenn, K.J., et al., *Investigating the effect of autoinduction in cynomolgus monkeys of a novel anticancer MDM2 antagonist, idasanutlin, and relevance to humans*. *Xenobiotica*, 2016. **46**(8): p. 667-76.
8. Andreeff, M., et al., *Results of the Phase I Trial of RG7112, a Small-Molecule MDM2 Antagonist in Leukemia*. *Clin Cancer Res*, 2016. **22**(4): p. 868-76.
9. Khoo, K.H., C.S. Verma, and D.P. Lane, *Drugging the p53 pathway: understanding the route to clinical efficacy*. *Nat Rev Drug Discov*, 2014. **13**(3): p. 217-36.
10. Ding, Q., et al., *Discovery of RG7388, a potent and selective p53-MDM2 inhibitor in clinical development*. *J Med Chem*, 2013. **56**(14): p. 5979-83.
11. Tovar, C., et al., *MDM2 small-molecule antagonist RG7112 activates p53 signaling and regresses human tumors in preclinical cancer models*. *Cancer Res*, 2013. **73**(8): p. 2587-97.
12. Vassilev, L.T., et al., *In vivo activation of the p53 pathway by small-molecule antagonists of MDM2*. *Science*, 2004. **303**(5659): p. 844-8.

13. Pan, R., et al., *Synthetic Lethality of Combined Bcl-2 Inhibition and p53 Activation in AML: Mechanisms and Superior Antileukemic Efficacy*. *Cancer Cell*, 2017. **32**(6): p. 748-760 e6.
14. Ishizawa, J., et al., *Mitochondrial Profiling of Acute Myeloid Leukemia in the Assessment of Response to Apoptosis Modulating Drugs*. *PLoS One*, 2015. **10**(9): p. e0138377.
15. Stewart, H.J., et al., *BRD4 associates with p53 in DNMT3A-mutated leukemia cells and is implicated in apoptosis by the bromodomain inhibitor JQ1*. *Cancer Med*, 2013. **2**(6): p. 826-35.
16. Conrad, R.J., et al., *The Short Isoform of BRD4 Promotes HIV-1 Latency by Engaging Repressive SWI/SNF Chromatin-Remodeling Complexes*. *Mol Cell*, 2017. **67**(6): p. 1001-1012 e6.
17. Wu, S.Y., et al., *Brd4 links chromatin targeting to HPV transcriptional silencing*. *Genes Dev*, 2006. **20**(17): p. 2383-96.

REVIEWERS' COMMENTS

Reviewer #1 (Remarks to the Author):

The authors have done some of the analyses that I have requested, however, in my opinion the quality of the work has not increased sufficiently to warrant publication in Nature Communications. The differences in impact on tumor cell growth in vitro and in vivo from the combination of an MDM2 inhibitor plus a BET inhibitor compared to treatment with either inhibitor are not impressive. I still argue that this is not impressive synergy.

Moreover, the authors have not rectified the issue that they are jumping in their different experiments between using two different MDM2 inhibitors. The reason for this is that if they had done the in vivo experiments with an MDM2 inhibitor that actually inhibits mouse MDM2 (they used one that ONLY binds to human MDM2), they would have seen on-target toxicity in healthy tissues. It is not fair to use an inhibitor that acts on only on the targeted protein in the tumor cells (human origin) but does not bind to the mouse protein in the healthy tissues.

Finally, I still believe that many of the data presented in this revised manuscript have already been presented in several previous publications from several groups. Thus, the novelty of the work presented is in my opinion not high.

Reviewer #2 (Remarks to the Author):

Having read through the rebuttal comments and the revised manuscript, the authors have satisfactorily answered and addressed my questions, and indeed those of my co-reviewer. In particular the demonstration of BRD4 as a repressor of TP53 genes is now much stronger and the preliminary mechanistic explanation of the recruitment of G9a (KMT1C), another potential therapeutic target, as a co-repressor is highly interesting. Whilst it would have been of further interest to the readership of Nat Comms to include this data in the current manuscript, I tend to agree with the authors that the data are somewhat preliminary and that it would be more responsibly presented separately in a more complete manuscript. I therefore now think that the manuscript is acceptable for publication.